

# Box model trajectory studies of contrail formation using a particle-based cloud microphysics scheme

Andreas Bier[1], Simon Unterstrasser[1], and Xavier Vancassel[2]

[1]Deutsches Zentrum für Luft- und Raumfahrt, Insitut für Physik der Atmosphäre, Oberpfaffenhofen, Germany
[2]ONERA, The French Aerospace Lab, Palaiseau, France

**Correspondence:** Andreas Bier (Andreas.Bier dlr.de)

**Abstract.**

We investigate the microphysics of contrail formation behind commercial aircraft by means of the particle-based LCM (Lagrangian Cloud Module) box model. We extend the original LCM to cover the basic pathway of contrail formation of soot particles being activated into liquid droplets that soon after freeze into ice crystals. In our particle-based microphysical approach, simulation particles are used to represent different particle types (soot, droplets, ice crystals) and properties (mass/radius, number). The box model is applied in two frameworks. In the classical framework, we prescribe the dilution along one average trajectory in a single box model run. In the second framework, we perform a large ensemble of box model runs using 25000 different trajectories inside an expanding exhaust jet as simulated by the LES (large-eddy simulation) model FLUDILES.

In the ensemble runs, we see a strong radial dependence of the temperature and relative humidity evolution. Droplet formation on soot particles happens first near the plume edge and a few tenths of seconds later in the plume centre. Averaging over the ensemble runs, the number of formed droplets/ice crystals increases more smoothly over time than for the single box model run with the average dilution.

Consistent with previous studies, contrail ice crystal number varies strongly with atmospheric parameters like temperature and relative humidity near the contrail formation threshold. Close to this threshold, the freezing fraction of soot particles depends strongly on the geometric-mean dry core radius and the hygroscopicity parameter of soot particles. This sensitivity is quite low at ambient conditions far away from the formation threshold. Absolute ice crystal numbers, on the other hand, are controlled by the soot number emission index for all atmospheric conditions.

The comparison with a recent contrail formation study by Lewellen (2020) (using similar microphysics) shows a later onset of our contrail formation due to a weaker prescribed plume dilution. If we use the same dilution data, our and Lewellen's evolution in contrail ice nucleation show an excellent agreement cross-validating both microphysics implementations. This means that differences in contrail properties mainly result from different representations of the plume mixing and not from the microphysical modelling.

The presented aerosol and microphysics scheme describing contrail formation is of intermediate complexity and thus suited to be incorporated in an LES model for 3D contrail formation studies explicitly simulating the jet expansion. The presented box model results will help interpreting the upcoming, more complex 3D results.



# 1 Introduction

Aviation contributes to around 5% of the anthropogenic climate impact (Lee et al., 2021), where contrail cirrus is the largest known contributor in terms of the radiative forcing (e.g., Boucher et al., 2013; Bock and Burkhardt, 2016). Due to projections

of large increases in air traffic, this contribution is expected to increase significantly (Bock and Burkhardt, 2019). The number of nucleated contrail ice crystals has a large impact on contrail cirrus properties and their life cycle (e.g., Unterstrasser and Gierens, 2010; Bier et al., 2017; Burkhardt et al., 2018). The formation of contrails depends on complex (thermo)dynamic, microphysical and chemical processes in the exhaust plume leading to a large variability in initial ice crystal numbers.

Measurements and observations implicate that contrails form once the plume mixture between the exhaust and ambient air

reaches or surpasses water-saturation. This condition is mathematically cast in form of the Schmidt-Appleman (SA)-criterion, a relation derived purely from the thermodynamics of the mixing process (Schumann, 1996).

Plume exhaust particles, in particular soot but also ambient particles entrained into the plume, can activate into water droplets and subsequently freeze to ice crystals by homogeneous nucleation (Kärcher et al., 2015). Ultrafine volatile particles resulting from recombination of chemi-ion clusters and molecules (Yu and Turco, 1997) may also contribute to the formation of ice

crystals but typically require huge water-supersaturation to form water droplets due to their small size of a few nanometers (Kärcher and Yu, 2009). Contrail ice nucleation typically occurs in the first second(s) of plume mixing during the jet phase.

Several studies on contrail formation in jet exhaust plumes have been conducted in the past. At the moment, there are basically two complementary model approaches to study the contrail formation which either focus on jet dynamics or on plume microphysics. On the one hand, 0D-box models based on a few idealised air parcel trajectories originating from an engine (e.g.,

Kärcher and Yu, 2009; Wong and Miake-Lye, 2010) have in general detailed microphysical and chemical schemes but neglect the large spatial variability in contrailing jet plumes. On the other hand, several three-dimensional (3D) large-eddy simulations (LES) studying the contrail formation in the near-field plume (Paoli et al., 2013; Khou et al., 2015, 2017) resolve many details of the jet flow and account for spatial variation in the plume. Even though they are equipped with sophisticated aerosol and chemical models, they have a simplified representation of microphysical processes. The main assumption in those 3D studies

is that contrail formation is triggered by heterogeneous ice nucleation on chemically activated soot particles following Kärcher (1998). Chemical activation means that at least 10% of the particle surface needs to be covered with sulphuric acid to form a thin liquid layer around the particle due to adsorption of water molecules. First, these studies assume water-saturation to be sufficient for the activation of soot particles, but the Kelvin effect requires, in particular for smaller soot particles, that water-saturation is clearly surpassed. This might cause a significant overestimation of the number of formed ice crystals. Second,

experimental investigations have shown that fresh engine soot particles are not supposed to be good ice nuclei (e.g., Bond et al., 2013). This strengthens the requirement of a transient liquid phase according to the SA-criterion. Third, aviation soot develops through incomplete combustion of hydrocarbons (Bockhorn, 1994) and is mainly composed of elementary carbon leading in general to a hydrophobic character. However, laboratory studies reveal the existence of active sites on soot parti-



cles, which develop by the attachment of functional groups and enable the adsorption of water molecules (Popovicheva et al.,
2017). Since nitrate species and hydroxyls have also a great potential to interact with soot particles (e.g., Kärcher et al., 1996;
Grimonprez et al., 2018), the coating of soot by sulphuric acid is not mandatory for their activation into water droplets. This
fact explains why visible contrails can still form when almost desulphurised fuel is burnt (Busen and Schumann, 1995).

Lewellen (2020) uses the more appropriate microphysical pathway consistent with the SA-criterion and considers soot, ambi-
ent and volatile particles as exhaust particles types potentially activating into water droplets and forming ice crystals. However,
these exhaust particles are assumed to have a monodisperse (or at best bidisperse) size distribution. This does not sufficiently
capture the Kelvin effect-related competition among differently-sized aerosol particles and possibly leads to too narrow ice
crystal size spectra. In general, Lewellen (2020) investigated the sensitivity of contrail ice crystal numbers against number
emission indices of soot, ambient aerosol and ultrafine particles and ambient conditions both within 3D LES and within a box
model.

The basic objective of the project *ConForm* funded by the Deutsche Forschungsgemeinschaft (DFG) is to combine the advan-
tages of the two complementary approaches, which means to simulate contrail formation in the expanding jet by means of 3D
LES with an improved representation of cloud microphysics. For this, we will employ the LES model EULAG (Smolarkiewicz
and Grell, 1992), which is fully coupled with the particle-based ice microphysics scheme LCM (Sölch and Kärcher, 2010). The
model system EULAG-LCM has been used for contrail simulations during the vortex phase (e.g., Unterstrasser, 2014) and the
dispersion phase (e.g., Unterstrasser et al., 2017a, b) in the recent past. So far, LCM includes ice crystal formation as it occurs
in natural cirrus clouds. The present paper describes how the LCM model has been extended in order to cover the specifics of
contrail formation and is a first step towards the goals of *ConForm*. Our main objective is to test the extended LCM in a simple
dynamical framework by running it in a box model setup. The microphysical parameterisation is similar to that in Lewellen
(2020) with the further improvement of prescribing soot particles by a lognormal size distribution. Moreover, the numerical
approach of the microphysics differs as our study relies on a particle-based description and not on an Eulerian spectral bin
model.

Paoli et al. (2008) and Vancassel et al. (2014) extended the box model framework to a multi-0D approach. Multi-0D here
means that the box model was run for a large ensemble of trajectories that sample the expanding jet plume as provided by
3D LES and that represent many different plume dilution evolution. In such an "offline" approach, the microphysics can be
more advanced than in a full 3D framework with "online" microphysics. In our study, we intend to highlight the importance
of the huge variability in thermodynamic and contrail properties resulting from the variability in the inhomogeneous turbulent
mixed jet plume. Therefore, we will use an ensemble of many trajectories where the data have been taken from Vancassel et al.
(2014). We expect that the temporal evolution of ensemble mean properties leads to improved scientific results compared to
those using one single average trajectory.

Paoli et al. (2008) mainly studied the temporal evolution of concentrations and (apparent) emission indices of volatile particles
and ice crystals depending on the fuel sulphur content. Vancassel et al. (2014) revealed the spatial variability in thermody-
namic quantities and contrail properties. Moreover, they used an online method with less detailed and an offline method with
more detailed contrail microphysics and highlighted differences between both methods. The two studies have the following





improvements compared to the online 3D studies of Khou et al. (2015, 2017): They include different types of aerosol particles,

the chemical processing of fuel sulphur and the formation and growth of volatile particles. Curvature and solution effect are accounted for. A further improvement is prescribing lognormally size-distributed soot particles using 60 size bins and varying the geometric mean dry core radius between 10 and 30 nm.

For conventional aircraft engines, soot particles are in general the major source for the formation of contrail ice crystals (e.g., Kärcher et al., 1996; Kärcher and Yu, 2009; Kleine et al., 2018). The number of emitted soot particles significantly influences

the contrail cirrus climate impact (Burkhardt et al., 2018). Since engine soot emissions are quite variable and uncertain particularly in terms of their number and size (e.g., Anderson et al., 1998; Petzold et al., 1999; Schumann et al., 2002; Agarwal et al., 2019), our main focus is to investigate the sensitivity of contrail properties (during their formation stage in the first seconds) on different soot properties. We will also analyse the variability of contrail ice nucleation with ambient conditions as already explored by Bier and Burkhardt (2019) using the analytical parameterisation of Kärcher et al. (2015).

## 2   Methods

This section gives a short overview of the basic "Lagrangian Cloud Module" (LCM)-model (Sect. 2.1), which has been extensively used for simulations of natural cirrus as well as young and aged contrails in the recent past. Section 2.2 then describes in more detail the modifications and extensions for simulating contrail formation and Sect. 2.3 gives an overview about the general plume evolution and our trajectory data. Finally, we will explain the box model framework, in which LCM is employed

in the present study.

### 2.1   Particle-based LCM microphysics

The particle-based ice microphysics model LCM (Sölch and Kärcher, 2010) comprises explicit aerosol and ice microphysics and is employed for the simulation of pure ice clouds like natural cirrus (e.g., Sölch and Kärcher, 2011) or contrails (e.g., Unterstrasser, 2014; Unterstrasser and Görsch, 2014; Unterstrasser et al., 2017a, b). In a particle-based microphysical approach,

hydrometeors (like ice crystals or water droplets) are described by simulation particles (SIPs) unlike traditional Eulerian approaches, where cloud properties are usually described by field variables. LCM has been used in conjunction with the LES flow solver EULAG, which computes the evolution of momentum, temperature and water vapour. We omit the details about the coupling between EULAG and LCM as in the present box model approach background temperature and water vapour concentration are prescribed and not simulated. Each SIP represents a certain number $\nu_i$ of real ice crystals with identical prop-

erties and stores information about the single ice crystal mass $m_i$, the weighting factor $\nu_i$, the ice crystal habit among others. Microphysical processes on the simulation particles include homogeneous freezing of liquid super-cooled aerosol particles, heterogeneous ice nucleation, deposition growth of ice crystals, sedimentation, aggregation, latent heat release and radiative impact on particle growth. For the contrail formation simulations, many of those processes (e. g., aggregation, sedimentation, radiation-related effects) are switched off. In the extended LCM, which will be described next, initial SIPs represent soot

particles, which then become activated into water droplets and eventually freeze (assuming suitable background conditions).



The SIP data structure is augmented and stores information about the particle type (soot, droplet, ice crystal) and properties (number, mass/radius, freezing temperature).

## 2.2 Microphysics of contrail formation

### 2.2.1 Exhaust particles

In this study, we consider soot as the only exhaust particle. Background particles will become relevant for aircraft with soot-poor or even soot-free emissions as for liquid hydrogen propulsion (Kärcher et al., 2015; Rojo et al., 2015). We neglect ultrafine volatile particles since this would require the inclusion of complex chemical processes like the recombination between ion-clusters leading to huge computational costs. These ultrafine particles are in particular relevant at ambient temperatures more than 10 K below the contrail formation threshold and for engines with soot-poor emissions still containing sulphuric and/or

organic compounds (Kärcher and Yu, 2009). The size spectrum of soot particles ranges from a few to hundreds of nanometer with geometric-mean values of around 15 nm (Petzold et al., 1999). Soot particles are composed of several nanometer-sized primary spherules combined in aggregates. For simplicity, we assume a spherical shape. Recent studies indicate a fractal irregular shape of the larger (more than several 10 nm) soot particles (e.g., Wang et al., 2019; Distaso et al., 2020). Taking this into account would modify their effective surface area and, therefore, the activation into water droplets and condensational

growth rates. But it would require additional complexity that may not be relevant in such an intermediate detailed microphysical approach.

We prescribe lognormally size distributed soot particles by an ensemble of $N_{SIP}$ simulation particles (SIPs) using a technique described in Appendix A of Unterstrasser and Sölch (2014). In a priory tests, we performed simulations with different values of $N_{SIP}$ ranging from 50 to 200. We find converged results of the analysed contrail properties for $N_{SIP}$ above around 100. Since

this threshold value slightly varies with the geometric-mean dry core radius and particle size distribution width, we choose a default value of around $N_{SIP} = 130$.

### 2.2.2 Basic formation pathway

In general, the moist exhaust plume cools due to mixing with ambient air. Contrails can form if ambient temperature is below the so called Schmidt-Appleman (SA)-threshold temperature ($\Theta_G$) and the plume becomes water-supersaturated in a certain

temperature range. The threshold temperature varies with atmospheric parameters (ambient relative humidity over water, air pressure) and engine/fuel properties (heat combustion, propulsion efficiency, water vapour emission index). The calculation of $\Theta_G$ is described in Appendix 2 of Schumann (1996).

We use the Kappa-Köhler theory (Sect. 2.2.3) to calculate which soot particles are able to activate into water droplets. Thereby, we prescribe the hygroscopicity parameter of soot particles as a measure for their solubility. This is a more convenient approach than defining a fixed fuel sulphur content since other polar species may lead to active sites on soot particles as well. Smaller

soot particles require higher water-supersaturation to overcome the Köhler barrier than larger ones (Fig. B1). Therefore, the latter particles can activate earlier and may suppress the droplet formation on smaller particles due to depletion of water vapour.





The super-cooled droplets subsequently grow by condensation as long as the exhaust air remains water-supersaturated. They freeze into contrail ice crystals when the homogeneous freezing temperature (Sect. 2.2.5) is reached. Larger droplets can freeze
earlier (i. e., at higher plume temperature) and grow further by deposition before smaller droplets manage to freeze into ice crystals.

### 2.2.3 Koehler theory

To calculate the equilibrium saturation ratio over a droplet or ice crystal surface ($S_K$), we use the Kappa-Köhler equation (Petters and Kreidenweis, 2007)

$$S_K = a_w \cdot \exp\left(\frac{Ke}{r}\right) = \frac{r^3 - r_d{}^3}{r^3 - r_d{}^3(1 - \kappa)} \cdot \exp\left(\frac{2\sigma M_w}{RT\rho_w r}\right), \tag{1}$$

consisting of the solution term, described by the activity of water $a_w$, and the exponential Kelvin term. $r_d$ is the particle dry core radius, $r$ the droplet radius and $\kappa$ the hygroscopicity parameter characterising the solubility of the aerosol particle; $\sigma$ is the surface tension of the solution droplet, $\rho_w$ the mass density of water and $T$ temperature. $M_w$ and $R$ denote the molar mass of water and the universal gas constant, respectively. For the activation of exhaust particles into water droplets, the plume
saturation ratio needs to overcome the maximum of $S_K$, denominated by the critical saturation ratio ($S_c$). This maximum increases with decreasing $r_d$ due to the Kelvin effect (Fig. B1 a)) and decreases with increasing $\kappa$ due to the solution effect (Fig. B1 b)). The calculation of $S_c$ is described in Appendix B.

The equilibrium saturation vapour pressures over the droplet (ice crystal) surface, used in Eqs. (2) and (7), are the product of the saturation vapour pressure over a flat water (ice) surface and $S_K$.

### 2.2.4 Condensational droplet growth

The droplet growth of activated soot particles is calculated according to Barrett and Clement (1988) and Kulmala (1993). In the steady state (small changes in the vapour flux) and for spherical droplets, the single droplet mass growth equation is given by

$$\frac{dm_w}{dt} = \frac{4\pi r (e_v - e_{K,w})}{\frac{R_v T}{D_v}\beta_m{}^{-1} + \frac{e_{K,w} L_c^2}{R_v \hat{K} T^2}\beta_t{}^{-1}} = 4\pi r \frac{e_v - e_{K,w}}{F_M \beta_m{}^{-1} + F_H \beta_t{}^{-1}}, \tag{2}$$

where $e_v$ is the partial vapour pressure and $e_{K,w}$ equilibrium saturation vapour pressure over the droplet surface. $L_c$ denotes the specific latent heat for condensation/evaporation, $D_v$ the binary diffusion coefficient of air and water vapour, $\hat{K}$ the conductivity of air and $R_v$ the specific gas constant of vapour. The terms $F_M$ and $F_H$ represent the mass and heat diffusion term with $\beta_m$ and $\beta_t$ denoting the transitional correction factors calculated according to Fuchs and Sutugin (1971).

### 2.2.5 Homogeneous freezing

We calculate the homogeneous freezing temperature ($T_{frz}$) of a super-cooled droplet with radius $r$ according to the method of Kärcher et al. (2015). This method is suitable for a strong cooling situation typically occurring in a jet plume. We introduce the





nucleation rate as the number of droplets freezing per unit time

$$j = J \cdot LWV, \tag{3}$$

where $J$ is the nucleation rate coefficient and $LWV = \frac{4}{3}\pi(r - r_d{}^3)$ is the liquid volume available for freezing. The temperature
dependent nucleation rate coefficient $J$ is parametrised according to Riechers et al. (2013)

$$J/(\mathrm{m^3 s}) = 10^6 \cdot \exp(a_1 T + a_2), \tag{4}$$

where $a_1 = 3.574\,\mathrm{K}^{-1}$ and $a_2 = 858.72$ are empirical constants. Assuming that the whole droplet freezes immediately if ice
nucleates within its volume, the pulse-like nature of $j$ may be expressed by a characteristic freezing time scale

$$\tau_\mathrm{frz}^{-1} = \frac{\partial \ln j}{\partial t} \simeq \frac{\partial \ln J}{\partial T}\dot{T} \tag{5}$$

with $\dot{T}$ denoting the plume cooling rate. Using Eq. (4) we obtain $\tau_\mathrm{frz} \simeq (a_1\dot{T})^{-1}$.

The homogeneous freezing requirement will be fulfilled if the product of $j = LWV \cdot J$ and $\tau_\mathrm{frz}$ is unity. Using this requirement,
inserting the relations from above and rearranging to $T := T_\mathrm{frz}$ yields

$$T_\mathrm{frz}(r,\dot{T}) = \frac{1}{a_1}\left[\ln\left(\frac{3 \cdot 10^{-6} a_1 \dot{T}}{4\pi\,(r^3 - r_d{}^3)} \cdot \mathrm{m^3 s}\right) - a_2\right]. \tag{6}$$

The homogeneous freezing temperature in contrails declines with decreasing droplet radius and increasing cooling rate. As
shown in Fig. 8 of Kärcher et al. (2015), $T_\mathrm{frz}$ ranges between around 230 and 232 K for typical droplet sizes (100–500 nm) at
cruise altitude conditions.

### 2.2.6 Depositional ice crystal growth

The mass change of a single ice crystal (Mason, 1971) is given by

$$\frac{dm_i}{dt} = \frac{4\pi\,C\,r\,D_v\beta_v{}^{-1}(e_v - e_{K,i})}{\frac{D_v\beta_v{}^{-1}L_d\,e_{K,i}}{\hat{K}\beta_k{}^{-1}\beta_v{}^{-1}T}\left(\frac{L_d}{R_v T} - 1\right) + R_v T}, \tag{7}$$

where $C$ is the shape factor, $r$ the equivalent volume radius and $L_d$ is the specific latent heat for deposition/sublimation. In our
study, we set C to unity assuming spherical ice crystals as they are typically observed for young contrails (e.g., Schröder et al.,
2000). Since sedimentation is of low relevance during contrail formation, we here neglect ventilation effects. The calculation
of the transitional correction factors $\beta_v$ and $\beta_k$ is described in Appendix A.

### 2.3 Plume evolution and trajectory data

In this section, we first explain the plume evolution assuming a mean state. Subsequently, we describe which kind of trajectory
data we will use and how dilution, a measure for the air-to-fuel ratio and the volume of the plume, is derived from these data.





### 2.3.1 General plume dilution equations

The temporal evolution of the mass mixing ratio of a species $\chi(t)$ is given by (Kärcher, 1995; Kärcher, 1999)

$$\dot{\chi}(t) = -\omega_\chi \left( \chi(t) - \chi_\mathrm{a} \right) + \xi(t), \tag{8}$$

where the first term describes the mixing of the hot exhaust with ambient air and the second term ($\xi$) represents sink and source terms. The entrainment rate $\omega_\chi$ characterises the dilution and $\chi_\mathrm{a}$ is the ambient mass mixing ratio of the species. Applying the equation above to water vapour and setting up an analogous equation for the evolution of the plume temperature ($T$) yields

$$\dot{x}_\mathrm{v}(t) = -\omega_\mathrm{v} \left( x_\mathrm{v}(t) - x_\mathrm{v,a} \right) + \dot{q}_\mathrm{PC}(t), \tag{9}$$
$$\dot{T}(t) = -\omega_\mathrm{T} \left( T(t) - T_\mathrm{a} \right) + \dot{T}_\mathrm{LH}(t), \tag{10}$$

where $T_\mathrm{a}$ and $x_\mathrm{v,a}$ are ambient temperature and vapour mass mixing ratio, respectively. The subscript $LH$ relates to latent heating and $PC$ stands for phase changes from water vapour to liquid water or ice and vice versa.

From the evolution of a passive tracer $\chi_\mathrm{p}(t)$ (defined by Eq. (8) with $\xi = 0$), we can derive the entrainment rate $\omega_\mathrm{p}$

$$\omega_\mathrm{p}(t) = -\frac{\dot{\chi}_\mathrm{p}(t)}{\chi_\mathrm{p}(t) - \chi_\mathrm{p,a}} \tag{11}$$

and set $\omega_\mathrm{v} = \omega_\mathrm{p}$ since water vapour is supposed to be diluted like a passive tracer. The entrainment rate characterising the
temperature change, $\omega_\mathrm{T}$, differs from $\omega_\mathrm{v}$ by a factor called the Lewis number $Le = \omega_\mathrm{v}/\omega_\mathrm{T}$, which is the ratio of the thermal and mass diffusivity. We set $Le$ to one assuming that heat and mass diffusion proceed equally fast since turbulent exchange processes are dominating over molecular exchange processes. Due to $\omega_\mathrm{v} = \omega_\mathrm{T}$, we skip the index from now on and simply use $\omega$. The entrainment rate $\omega$ describes the instantaneous mixing strength. Integrating it over time, we obtain the plume dilution factor (Kärcher, 1995)

$$\mathcal{D}(t) = \exp\left( -\int_{t_0}^{t} \omega(t') dt' \right) = \frac{\chi_\mathrm{p}(t) - \chi_\mathrm{p,a}}{\chi_\mathrm{p,0} - \chi_\mathrm{p,a}}, \tag{12}$$

where the subscript "0" refers to conditions at the engine exit at time $t_0 := 0\,\mathrm{s}$.

At the engine exit, the emitted air is already a mixture of ambient air and combustion products. The initial plume dilution $\hat{N}_0$ can be derived from the temperature excess $T_0 - T_\mathrm{a}$

$$\hat{N}_0 = \frac{Q\,(1-\eta)}{\bar{c}_\mathrm{p}\,(T_0 - T_\mathrm{a})}, \tag{13}$$

where $Q$ is the specific heat of combustion, $\eta$ the propulsion efficiency and $\bar{c}_p = 1020\,\mathrm{J(kg\,K)}^{-1}$ an average heat capacity for dry air over a temperature range between around 200 and 600 K (see derivation in Schumann, 1996).

The overall dilution $\hat{N}$ generally increases with plume age due to continuous entrainment of ambient air and is related to the dilution factor (this quantity decreases with plume age) via

$$\hat{N}(t) = \hat{N}_0/\mathcal{D}(t), \tag{14}$$




implying $\mathcal{D}_0 = 1$.

The initial plume area $A_0$ results from mass conservation (Schumann et al., 1998)

$$A_0 = \frac{\hat{N}_0 \, m_{\mathrm{F}}}{\rho_0}, \tag{15}$$

where $\rho_0 = \frac{p_{\mathrm{a}}}{R_{\mathrm{d}} T_0}$ is the air density with $p_{\mathrm{a}}$ denoting air pressure, $R_{\mathrm{d}}$ the specific gas constant of dry air and $m_{\mathrm{F}}$ the fuel consumption (in $\mathrm{kg \, m^{-1}}$). The evolution of the plume area is given by

$$A(t) = A_0 \cdot \frac{\rho_0}{\rho(t) \, \mathcal{D}(t)} = A_0 \cdot \frac{T(t)}{T_0 \, \mathcal{D}(t)}. \tag{16}$$

In the numerical implementation, the ordinary differential equations (ODE) like Eqs. (9) and (10) are discretised by a simple Euler forward scheme. Due to the linearity of the ODEs, the implementation of an implicit Euler backward or a second-order trapezoidal scheme is straight-forward, but we find no improvement over the forward scheme. The source terms in the ODEs are provided by the LCM. In the LCM, the computation involves summations over all SIPs with phase changes or water vapour

uptake/release during the time step. All model components (plume evolution ODE and LCM microphysics) use a time step of 0.001 s. This time step is sufficiently small to account for the fast diffusional growth at high plume supersaturations.

### 2.3.2   Trajectory ensemble data

Vancassel et al. (2014) used the LES solver FLUDILES to simulate the plume evolution in a 3D domain over 10 s starting behind the engine nozzle of an A340-300 aircraft. They sampled the plume with 25000 trajectories that were randomly posi-

tioned inside the initial plume area $A_0 = \pi r_0^2$ with radius $r_0 = 0.5\,\mathrm{m}$. We assume that all trajectories represent an equal share of the plume. For each trajectory, the temperature evolution $T_{\mathrm{3D},k}(t)$ has been tracked and we use these data to infer the dilution factor (or equivalently the entrainment rate) along each trajectory $k$ by assuming that temperature is a passive tracer:

$$\mathcal{D}_k(t) = \frac{T_{\mathrm{3D},k}(t) - T_{\mathrm{3D,a}}}{T_{\mathrm{3D},0} - T_{\mathrm{3D,a}}}. \tag{17}$$

$T_{\mathrm{3D},0} = 580\,\mathrm{K}$ and $T_{\mathrm{3D,a}} = 220\,\mathrm{K}$ are the plume exit and ambient temperature of the FLUDILES simulation. These equations

are only meaningful, if $T_{\mathrm{3D},k} > T_{\mathrm{3D,a}}$ is fulfilled. However, the LES temperature occasionally falls below $T_{\mathrm{3D,a}}$ due to a pressure drop resulting from the interaction of the plume with the wake vortex. In the computation of $\omega_k(t)$ or $\mathcal{D}_k(t)$, we replace all values of $T_{\mathrm{3D},k}$ being below $T_{\mathrm{3D,a}} + \epsilon$ with this lower limit. The threshold $\epsilon$ is chosen such that the maximum dilution is still reasonable for a few seconds of plume age ($\mathcal{D}_{\min} \approx 2.5 \cdot 10^{-3}$).

In the FLUDILES simulation, the initial plume temperature profile is based on the Crocco-Busemann relation (White, 1974)

and introduces a smooth radial transition between the plume and the environment for numerical reasons. This means that $T_{\mathrm{3D},k}(t_0)$ equals $T_{\mathrm{3D},0}$ only directly behind the nozzle exit centre and decreases down to around $400\,\mathrm{K}$ towards the plume edge (see later in Fig. 1 a)). These trajectories start with $\mathcal{D}_{k,0} < 1$ and an accordingly higher initial dilution $\hat{N}_{k,0}$. For simplicity, we mostly neglect this fact in the following description, but the initialisation in this transition region is consistently treated in our model.





The process of converting jet kinetic energy into thermal energy is considered within the 3D simulations. This means that the FLUDILES temperature is higher than a "passive tracer temperature". Therefore, we underestimate dilution causing a later onset of contrail formation. This effect is partially compensated by the inhomogeneous initial temperature profile implying that exhaust air parcels are already diluted near the plume edge.

Basically, we only infer the dilution from the trajectory data according to Eq. (17) and plug the corresponding entrainment
rate values into Eqs. (9) and (10) for each trajectory. For the radial plots presented later on, the coordinate data of these trajectories are used, but the actual computations are independent of this type of information.

Note that the ambient temperature $T_\mathrm{a}$ and plume temperature at engine exit $T_0$ in Eqs. (9) and (10) are allowed to differ from $T_\mathrm{3D,a}$ and $T_\mathrm{3D,0}$ used in Eq. (17). Similarly, the ambient and initial plume water vapour can be chosen independently of the 3D data. In such a case, we basically use only the information of the dilution and its variability.

Moreover, a single mean state trajectory can be derived from the trajectory ensemble. We determine the corresponding mean dilution $\mathcal{D}_\mathrm{AT}$ by averaging the pre-processed temperature data over all 25000 trajectories and plugging the average temperature evolution $\bar{T}_\mathrm{3D}$ into Eq. (17).

## 2.4    Box model framework

We use the LCM box model, which is based on the 3D-LCM (introduced in Sect. 2.1) and has been extended for the contrail
formation microphysics. Similar to Paoli et al. (2008), we either perform a single box model run with an average dilution (labelled "average traj") or an ensemble of box model runs for the trajectory dilution data. In this case, physical quantities are suitably averaged in the end (labelled "ensemble mean"). The simulated time $t_\mathrm{sim}$ is 2 s. In the following, we describe the initial conditions (Sect. 2.4.1) and introduce important diagnostics (Sect. 2.4.2).

### 2.4.1    Initial conditions

Table 1 summarises initial conditions of our baseline scenario, which are described in the following. The plume exit temperature (near the plume centre) is set to 580 K. We prescribe an ambient temperature of either 220 K (baseline case) or 225 K (near-threshold case). The baseline SA-threshold temperature ($\Theta_\mathrm{G}$) is 226.2 K for the given atmospheric conditions and engine/fuel properties. The prescribed ambient relative humidity over ice, $RH_\mathrm{ice,a}$, determines the ambient water vapour mixing ratio $x_\mathrm{v,a}$ being proportional to $RH_\mathrm{ice,a} \cdot e_\mathrm{s,ice}(T_\mathrm{a})$. We use $RH_\mathrm{ice,a} = 120\%$ for the baseline case.

Engine and fuel parameters like plume exit temperature, propulsion efficiency and water vapour mass emission index have values according to an A340-300 aircraft at cruise speed (Table 1). This has been chosen consistently with the trajectory data setup. Only for the fuel consumption we prescribe a lower value than typical for this aircraft type in order to be consistent with the initial plume area given by the trajectory data. For given $T_0$, combustion heat and propulsion efficiency, we obtain $\hat{N}_0 \approx 75$ according to Eq. (13). The initial plume water vapour mixing ratio is controlled by the water vapour emission index $EI_\mathrm{v}$

$$x_\mathrm{v,k}(t_0) = \frac{EI_\mathrm{v}}{\hat{N}_\mathrm{0,k}} + \frac{\hat{N}_\mathrm{0,k} - 1}{\hat{N}_\mathrm{0,k}} x_\mathrm{v,a}, \tag{18}$$





where the second term is the contribution of ambient humidity entering and exiting the engine. This term is less than 1% of the total plume water vapour and, therefore, neglected in our initialisation.

Based on laboratory measurements, we prescribe lognormally distributed soot particles with a geometric-mean dry core radius ($\bar{r}_d$) of 15 nm and geometric width of 1.6 (Hagen et al., 1992; Petzold et al., 1999) and set the engine soot number

emission index ($EI_s$) to $10^{15}\,\mathrm{kg}^{-1}$ as a typical value for commercial aircraft (e.g., Schumann et al., 2002, 2013; Kleine et al., 2018). Furthermore, we assume the same hygroscopicity parameter value ($\kappa_s = 0.005$) as in Kärcher et al. (2015). The overall number of emitted engine soot particles per flight distance is given by

$$N_{s,tot} = EI_s \cdot m_F. \tag{19}$$

Table 2 summarises the parameter variations (PV) that investigate the sensitivity of contrail formation to soot properties

(PV1-PV4) and ambient conditions (PV5/PV6). In the first two PVs, we either vary $EI_s$ or $\bar{r}_d$. Flight measurements have shown that a 50:50 mixing of conventional kerosene with a biofuel causes a decrease in $EI_s$ by around 50% (Moore et al., 2017). Additionally, ground measurements with synthetic Fischer-Tropsch fuels have found a decrease of the average soot particle size besides the decreased soot particle number (Beyersdorf et al., 2014). This is because the switch to alternative fuels leads to a reduced number of primary soot spherules. The subsequent coagulation between those spherules decreases and

the final soot aggregates are on average smaller. Therefore, in PV3, we vary both $EI_s$ and $\bar{r}_d$ and combine a 50% reduction in $EI_s$ with a reduction in $\bar{r}_d$ to 12.5 nm. In PV4, we change the hygroscopicity parameter of soot particles in a reasonable range between 0.001 and 0.01 (pers. communication with M. Petters). Finally, we study the variation of contrail properties with ambient temperature $T_a$ (PV5) and relative humidity over ice $RH_{ice,a}$ (PV6).

### 2.4.2  Diagnostics

Here, we introduce diagnostics that are used to evaluate the simulations.

The fraction of soot particles activated into water droplets ($\phi_{act}$) or frozen to ice crystals ($\phi_{frz}$) is given by

$$\phi_{act/frz} = \frac{\sum_{i=1}^{N_{SIP}} \zeta_{i,act/frz} N_{s,i}}{N_{s,tot}}, \tag{20}$$

where $\zeta_i$ characterises, whether soot particles ($\zeta_{i,act} = \zeta_{i,frz} = 0$) represented by a SIP $i$ have turned into water droplets ($\zeta_{i,act} = 1$

and $\zeta_{i,frz} = 0$) or ice crystals ($\zeta_{i,act} = \zeta_{i,frz} = 1$). Note that we classify ice crystals as activated, hence we get $0 \leq \phi_{frz} \leq \phi_{act} \leq 1$.

We introduce the apparent ice number emission index, $AEI_{ice}$, as a measure of the number of formed ice crystals per fuel mass burnt. Since we consider only soot as exhaust particles in our study, $AEI_{ice}$ is simply given by the product of $\phi_{frz}$ and $EI_s$.

Furthermore, we define the effective radius of the soot/droplet/ice number distribution

$$r_{eff} = \frac{\sum_{i=1}^{N_{SIP}} r_i^3 \cdot N_{s,i}}{\sum_{i=1}^{N_{SIP}} r_i^2 \cdot N_{s,i}}. \tag{21}$$

Typically, $r_{eff}$ is a quantity that is relevant in radiation problems. Here, we prefer $r_{eff}$ over the ordinary mean radius since the time evolution is smoother and there is no need to specify the total particle number.





| ambient conditions | fuel/engine properties | engine exit conditions | soot particle properties | numerical parameters |
|---|---|---|---|---|
| $T_\mathrm{a} = 220\,\mathrm{K}$ | $EI_\mathrm{v} = 1.26\,\mathrm{kg\,kg^{-1}}$ | $T_0 = 580\,\mathrm{K}$ | $EI_\mathrm{s} = 10^{15}\,\mathrm{kg^{-1}}$ | $t_\mathrm{sim} = 2\,\mathrm{s}$ |
| $p_\mathrm{a} = 240\,\mathrm{hPa}$ | $Q = 4.3 \cdot 10^7\,\mathrm{J\,kg^{-1}}$ | $\hat{N}_0 \approx 75$ | $\bar{r}_\mathrm{d} = 15\,\mathrm{nm}$ | $dt = 0.001\,\mathrm{s}$ |
| $RH_\mathrm{ice,a} = 120\%$ | $\eta = 0.36$ | $A_0 = 0.25\pi\,\mathrm{m^2}$ | $\kappa_\mathrm{s} = 0.005$ | $N_\mathrm{SIP} = 130$ |
| $T_\mathrm{a} = 218.8\,\mathrm{K}$ | $EI_\mathrm{v} = 1.25\,\mathrm{kg\,kg^{-1}}$ | $T_0 \approx 500\,\mathrm{K}$ | $EI_\mathrm{s} = 10^{16}\,\mathrm{kg^{-1}}$ | $t_\mathrm{sim} = 1\,\mathrm{s}$ |
| $p_\mathrm{a} = 238.4\,\mathrm{hPa}$ | $Q = 4.29 \cdot 10^7\,\mathrm{J\,kg^{-1}}$ | $\hat{N}_0 \approx 92$ | Mono20 or Bi10/40 | $dt = 0.001\,\mathrm{s}$ |
| $RH_\mathrm{ice,a} = 110\%$ | $\eta = 0.325$ | $A_0 = 0.31\,\mathrm{m^2}$ | $\kappa_\mathrm{s} = 0.005$ | $N_\mathrm{SIP} = 130$ |

**Table 1.** Baseline parameters of all regular LCM simulations (first 3 rows) and those corresponding to the box model set-up of Lewellen (2020) (last 3 rows). Hereby, Mono20 stands for a monodisperse distribution with $r_\mathrm{d} = 20\,\mathrm{nm}$ and Bi10/40 for a bidisperse distribution with a 50/50 mixture of 10 and 40 nm particles.

| PV1: $EI_\mathrm{s}/EI_\mathrm{s,ref}$ | PV2: $\bar{r}_\mathrm{d}$ / nm | PV3: $EI_\mathrm{s}/EI_\mathrm{s,ref}$ and $\bar{r}_\mathrm{d}$ | PV4: $\kappa_s$ | PV5: $T_\mathrm{a}$ / K | PV6: $RH_\mathrm{ice,a}$ / % |
|---|---|---|---|---|---|
| 0.5 | 12.5 | 0.5 and 12.5 | 0.001 | 215 | 100 |
| **1.0** | **15.0** | **1.0 and 15.0** | **0.005** | **220** | **120** |
| 2.0 | 20.0 | 2.0 and 20.0 | 0.01 | 225 | 140 |

**Table 2.** Parameter variations ("PV") of various soot properties and ambient conditions. The values in bold depict the default baseline parameter and $EI_\mathrm{s,ref}$ is the baseline soot number emission index.

# 3 Results

In this section, we first describe the spatio-temporal evolution of thermodynamic and contrail properties during the formation stage (Sect. 3.1). We then analyse the temporal evolution of contrail properties depending on different soot properties and atmospheric parameters in Sect. 3.2. Finally, we summarise our results by studying the sensitivity of final contrail ice crystal numbers to ambient temperature and relevant soot properties. Our simulations are based on the FLUDILES dilution data and we particularly compare simulations using the full trajectory ensemble and a single average trajectory.

## 3.1 Spatial variation of contrail properties

The first two rows of Fig. 1 depict the spatio-temporal evolution of temperature and relative humidity. The panels on the right show contour plots of $RH_\mathrm{liq}$ in $(r,t)$-space. The size of the exhaust plume is immediately apparent from this representation. The jet plume expands from an initial radius of 0.5 m towards around 7.5 m after $t = 1\,\mathrm{s}$. The black line indicates the plume radius $r_\mathrm{p}$ as a function of time. All other panels show line plots, where the horizontal axis uses the normalised radial distance $\tilde{r} = r/r_\mathrm{p}$ as coordinate. Panel a) shows the radial temperature distribution for different plume ages, as indicated in the inserted legend. The black line shows the initial temperature profile with the smoothing in the outer areas according to the Crocco-





Busemann profile. In the very beginning, the plume centre is not affected by entrainment and the core temperature remains at its initial value. The radial temperature gradient is initially large (nearly a temperature decrease of 300 K at $t = 0.05$ s) and decreases with increasing plume age as the plume centre cools. Later on, the absolute changes in temperature, spatially and temporally, decrease continuously.

Panel b) shows the radial $RH_{\mathrm{liq}}$ distribution of a simulation with switched-off contrail microphysics, where water vapour behaves like a passive tracer. We call the displayed quantity hypothetical $RH_{\mathrm{liq}}$; it increases with increasing plume age and decreasing plume temperature due to the non-linearity of the saturation vapour pressure. Since plume cooling starts earlier and proceeds faster near the plume edge, water-saturation is surpassed earlier in this region (after 0.25 s) than in the plume centre (after 0.5 s). For fixed plume age, the hypothetical $RH_{\mathrm{liq}}$ increases with increasing radial distance. This becomes also apparent

from panel c), where the radial dependence of the hypothetical $RH_{\mathrm{liq}}$ is depicted for several points in time.

Panels d), e) and f) show results of a simulation with switched-on contrail microphysics. As soon as droplet formation on soot particles is initiated (i.e., after plume relative humidity exceeds at least water-saturation), the further increase in $RH_{\mathrm{liq}}$ is inhibited due to condensational loss of water vapour. The subsequent rapid decrease in $RH_{\mathrm{liq}}$ (first at the plume edge and later on in the plume centre) is due to ice crystal formation and subsequent depositional growth. After around 1 s, the relative

humidity with respect to ice reaches values of 100%, which translates into water-subsaturated plume conditions. Panel e) shows the fraction of activated soot particles ($\phi_{\mathrm{act}}$). According to the evolution in plume relative humidity, soot particles activate into water droplets first near the plume edge and afterwards in the plume centre. Water saturation in the plume is first reached for $T_{\mathrm{sat}} \approx 240$ K so that below those temperatures soot particles may form water droplets and grow further by condensation. The homogeneous freezing of the super-cooled water droplets typically occurs at $T_{\mathrm{frz}} \approx 230 - 232$ K depending on the droplet size

and the cooling rate. Panel f) shows the fraction of soot particles that turned into ice crystals ($\phi_{\mathrm{frz}}$). The radial profiles of $\phi_{\mathrm{frz}}$ behave similar to the corresponding profiles of $\phi_{\mathrm{act}}$ but with a time lag of around 0.2 s. In the outer part with $\tilde{r} > 0.5$, basically all soot particles have become ice crystals after around 0.6 s. At $t = 0.6 - 0.8$ s, the profiles exhibit local minima at $\tilde{r} \approx 0.2$, which results from the minimum in $RH_{\mathrm{liq}}$ after around 0.7 s (Fig. 1 b).

### 3.2 Temporal evolution of contrail properties

#### 3.2.1 Mean state trajectory versus trajectory ensemble

We employ the box model in two different frameworks as outlined in Sect. 2.4. In the average traj framework, a single box model run with an average dilution is performed. In the ensemble mean framework, box model runs are carried out for the complete set of 25000 trajectories.

Fig. 2 compares the plume relative humidity (top row) and the activation/freezing fraction between both frameworks. This

is done for baseline conditions (left column, $T_a = 220$ K) and a near-threshold case (right column, $T_a = 225$ K with $|T_a - \Theta_G| \approx 1.5$ K). First, we analyse the baseline case. For the average traj, $RH_{\mathrm{liq}}$ surpasses water-saturation after around 0.4 s and leads immediately to a rapid droplet formation on soot particles. Thereby, $\phi_{\mathrm{act}}$ increases quasi pulse-like from zero to its final value. Subsequently, $RH_{\mathrm{liq}}$ decreases and approaches 100%. After 0.7 s the droplets freeze rapidly into ice crystals and the





**Figure 1.** Radial profiles of a) temperature and b), c), d) relative humidity over water $RH_{liq}$ in %, e) fraction of soot particles activating into water droplets $\phi_{act}$ and f) freezing to ice crystals $\phi_{frz}$. Whereas panels a), b), c) show results from a simulation with switched-off contrail microphysics, the three other panels show results from the baseline case with switched-on contrail microphysics. The contour plots in b) and d) show $RH_{liq}$ over $(r, t)$-space where $r$ is the absolute radial distance from the plume centre. The black line in b) indicates the plume radius $r_p$ over time. The line plots in the four other panels use the normalised radial distance $r/r_p$ as horizontal coordinate.





depositional ice crystal growth causes a steeper decrease in $RH_{liq}$ until ice-saturation is reached (see blue line in panel a)).

Regarding the ensemble mean evolution, the fraction of soot particles forming droplets/ice crystals increases smoothly over a certain time window. This is explained as follows: The spatial variability in $RH_{liq}$ is quite large (Fig. 1 d)) due to a systematic radial gradient and superposed turbulent deviations. Therefore, droplet and ice crystal formation occur for different trajectories at a different plume age mostly depending on the radial distance to the plume centre (Fig. 1 e) and f)). Although the ensemble mean features a smaller $RH$ peak value than the average traj evolution, a larger fraction of soot particles forms ice crystals

(96% vs. 88%).

In the near-threshold case (right column), $T_{sat} \approx 236$ K is lower and closer to $T_a$. Hence, water-saturation is reached at a higher plume age when the cooling rates are already lower. Additionally, the peak $RH$ values are clearly reduced. According to the evolution of the plume relative humidity, droplet and ice crystal formation on soot particles are initiated a few tenths of seconds later compared to $T_a = 220$ K (panel d)). The time between the activation of the first droplets and the freezing of the last droplets

is also longer (1.6 s versus 1.2 s). The final $\phi_{act}$ and $\phi_{frz}$-values are clearly reduced relative to the baseline case since only the larger soot particles can activate into water droplets due to the decreased peak $RH_{liq}$. Again, the ensemble mean evolution displays slightly higher final $\phi_{frz}$-values than the average traj.

A closer look at the average traj evolution of the near threshold case reveals that $\phi_{act}$ is not a monotonically increasing function and that the final $\phi_{frz}$-value is lower than the $\phi_{act}$-value at some intermediate time (in this example at $t = 0.9$ s). This is explained

as follows: The largest soot particles are activated first and over time smaller soot particles grow large enough to be considered activated as well. However soon after, $RH_{liq}$ drops below $S_K$ (for the smaller soot particles) as condensational growth removes water vapour. Those particles shrink, such that their radius drops below the activation radius. Once the largest water droplets freeze, depositional growth leads to a fast depletion of water vapour. Again, the smallest of the currently activated droplets face water-subsaturated conditions and also become de-activated (at around $t = 1.25$ s). We will encounter this "deactivation

phenomenon" later again. Even though it is not visible in the ensemble mean evolution, this deactivation phenomenon also occurs for most trajectories of the ensemble runs.



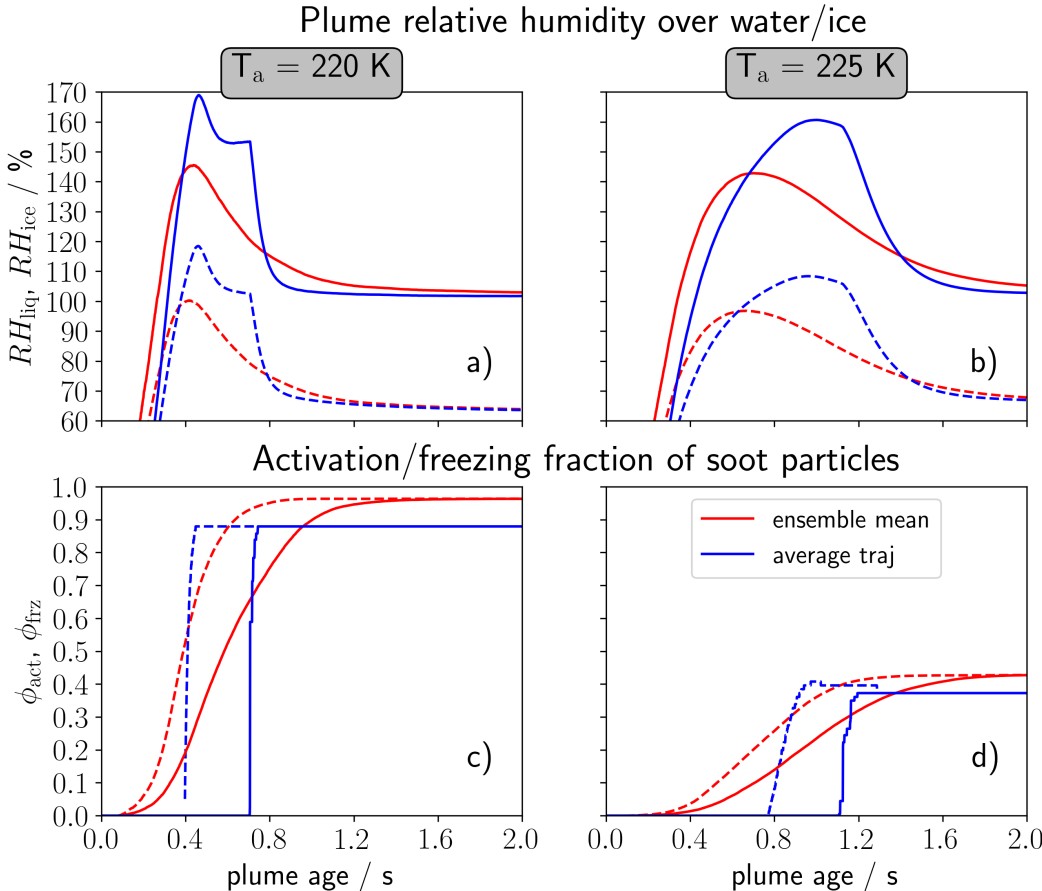

**Figure 2.** Temporal evolution of (top row) plume relative humidity over water (dashed) and ice (solid) and of (bottom row) fraction of soot particles activating into water droplets (dashed) and forming ice crystals (solid). The left column shows the baseline case with ambient temperature $T_a = 220\,\mathrm{K}$, the right column the near-threshold case with $T_a = 225\,\mathrm{K}$. Results are shown for the ensemble mean (red) and the average traj (blue) evolution.





### 3.2.2 Variation with soot properties

In this section, we investigate the sensitivity of contrail formation to various soot properties. Fig. 3 presents results of the sensitivity studies PV1 to PV4. As in the previous section, the baseline and near-threshold temperature cases are considered.

We first analyse the baseline temperature case ($T_a = 220\,K$; left column in Fig. 3). The ensemble mean results reveal that the temporal evolution of the activation fraction $\phi_{act}$ and freezing fraction $\phi_{frz}$ hardly changes with any variation of the soot properties. In the average traj framework, the sensitivity to soot properties is slightly higher. There, activation and freezing fraction moderately increase with decreasing $EI_s$ (panel a)) and increasing $\bar{r}_d$ (panel b)). A higher number of soot particles leads to a faster depletion of plume relative humidity prohibiting the activation of smaller particles. Likewise, a decrease of $\bar{r}_d$

reduces the number of soot particles that manage to be activated. Changing the two parameters, $EI_s$ and $\bar{r}_d$, simultaneously (panel c)), the effects on $\phi_{act}$ and $\phi_{frz}$ cancel out more or less resulting in an almost negligible effect. The variation of $\phi_{frz}$ with the hygroscopicity of soot particles (panel d) displays the lowest impact within the considered parameter range. Note that the average traj simulations with higher prescribed $EI_s$ (dotted cyan lines) exhibit the aforementioned deactivation phenomenon.

In the near-threshold scenario, the $\phi_{act}$ and $\phi_{frz}$-values are clearly lower due to decreased plume water-supersaturation

(Sect. 3.2.1). While the variation with $EI_s$ (panel b)) is still low, both average traj and the ensemble mean evolution are more sensitive to the variation of $\bar{r}_d$ and $\kappa$ than for $T_a = 220\,K$. This is because of two basic reasons: First, the critical saturation ratio for the activation of soot particles varies non-linearly with the soot core size (due to the non-linear Kelvin term) displaying a strong change for $r_d < 10\,m$ (Fig. B1 a)). This means that far away from the formation threshold, larger changes of $\bar{r}_d$ and $\kappa$ are necessary to affect the activation of the several nanometer sized soot particles. Second, these small soot particles

are at the tail of the size-distribution and, therefore, their activation makes only a low contribution to the overall $\phi_{act}$. In the near-threshold case, only the larger soot particles ($r_d \gtrsim 20\,nm$) can form water droplets. This activation threshold is close to $\bar{r}_d$ which simultaneously defines the maximum of the lognormal size distribution. Hence, changing $\bar{r}_d$ or $\kappa$ has a significantly higher impact on $\phi_{act}$ for $T_a = 225\,K$ than for lower ambient temperature.

Again, we find a stronger sensitivity to $r_d$ (panel b)) than for $\kappa$ (panel g)). Interestingly, the ensemble mean simulations show

in the end larger $r_d$ and $\kappa$-induced variations in $\phi_{frz}$ than the average traj simulations, which is opposite to what we saw in panel g). This is mainly due to the deactivation phenomenon. In the combined variation of $EI_s$ and $\bar{r}_d$ (panel f)), the impact of $\bar{r}_d$ on $\phi_{act}$ and $\phi_{frz}$ dominates over that of $EI_s$. Halving the $EI_s$-value and decreasing $\bar{r}_d$ to 12.5 nm leads to a decrease in $\phi_{frz}$ from 0.41 to 0.31 and translates into a decrease of $AEI_{ice}$ even by 62% (instead of around 50% if only the $EI_s$-value is halved).

Furthermore, we have investigated the sensitivity of contrail formation to the geometric width of the soot particle size distribu-

tion $\sigma$ (not shown): In contrast to the variation with $\bar{r}_d$ and $\kappa$, there is a negligible impact near the contrail formation threshold ($T_a = 225\,K$). This is because the activation of soot particles with dry core radii below $\bar{r}_d$ is already inhibited due to the Kelvin effect. For our baseline case, in general a higher fraction of soot particles forms ice crystals if the geometric width is reduced. This is due to the fact that the smallest soot particles within the size distribution become larger and hence can be activated more easily. Decreasing (increasing) geometric width from 1.6 to 0.8 (2.4) causes a decrease (increase) in $\phi_{frz}$ from 0.88 to

0.83 (0.98) for the average traj and from 0.96 to 0.93 (0.99) for the ensemble mean.

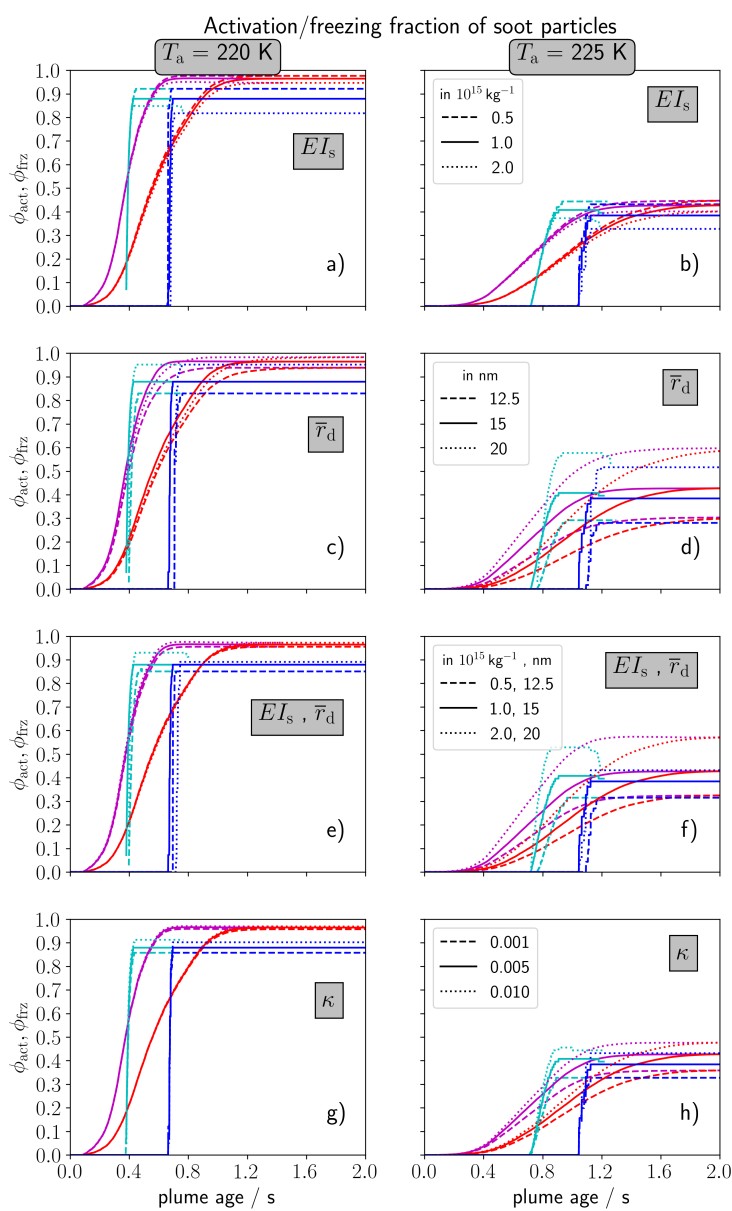

**Figure 3.** Temporal evolution of the fraction of soot particles activating into water droplets $\phi_{\text{act}}$ (cyan/magenta) and forming ice crystals $\phi_{\text{frz}}$ (blue/red) for a variation of the soot number emission index $EI_{\text{s}}$ (first row), geometric-mean dry core radius $\bar{r}_{\text{d}}$ (second row), a combined variation of $EI_{\text{s}}$ and $\bar{r}_{\text{d}}$ (third row) and a variation of the hygroscopicity parameter $\kappa$ (last row). The left column shows the baseline ($T_{\text{a}} = 220$ K) and the right column a near-threshold case ($T_{\text{a}} = 225$ K). The magenta/red lines show the ensemble mean and the cyan/blue lines the average traj results. The different line types (see legends on the right column) represent the according soot parameter variations (PV1-PV4).





### 3.2.3 Sensitivity to ambient conditions

In this subsection, we study the dependence of contrail formation on ambient conditions. We vary the ambient temperature $T_a$ and background relative humidity over ice $RH_{ice,a}$ (see PV5 and PV6 in Table 2).

The top row of Fig. 4 displays the variation of ambient temperature $T_a$ at default $RH_{ice,a} = 120\%$. As already explained above, a change of $T_a$ affects the contrail formation time and the number of soot particles turning into ice crystals. The lower $T_a$, the earlier water saturation is first reached in the plume and the higher is the attained peak water-supersaturation. Hence, reducing $T_a$ to 215 K, droplet and ice crystal formation are initiated earlier and more ice crystals form (panel a)). The results for $T_a = 220$ K and 225 K have already been described in Sect. 3.2.2. Thereby, the difference in $\phi_{frz}$ is significantly larger between 225 K (dotted) and 220 K (solid) than that between 220 and 215 K (dashed).

The evolution of the effective radius $r_{eff}$ in panel b) shows a steep increase from the moment on when the first droplets are activated. The increase slows down in most cases and is accelerated again, once the droplets freeze. The curves level off as soon as ice-saturation is reached. This happens after around 1 s for the lower $T_a$ and after around 2 s for $T_a = 225$ K. The final $r_{eff}$ values hardly change between $T_a = 215$ K and 220 K and between the average traj and the ensemble mean. Only for $T_a = 225$ K higher values are attained. This is because the absolute amount of ambient water vapour is higher for higher $T_a$ and gets distributed among fewer ice crystals.

The bottom row of Fig. 4 shows the variation of $RH_{ice,a}$. For $T_a = 220$ K (panel c)), contrail ice nucleation is not at all affected by the variation of $RH_{ice,a}$. In contrast, the sensitivity of $\phi_{frz}$ to $RH_{ice,a}$ is very high for $T_a = 225$ K. At ice-saturation (dotted line), only few ice crystals form ($\phi_{frz} < 10\%$). This is because a decrease in $RH_{ice,a}$ causes a decrease in $\Theta_G$ from 226.2 to 225.5 K so that those contrails form only 0.5 K below the SA-threshold temperature and maximum plume water-supersaturation is only a few percent. Analogously, more ice crystals form for $RH_{ice,a} = 140\%$ as $\Theta_G$ and hence $|T_a - \Theta_G| \approx 3$ K is higher.

Our results highlight that the sensitivity of contrail ice nucleation to atmospheric parameters is only large near the contrail formation threshold which is consistent with previous studies (e.g., Kärcher and Yu, 2009; Kärcher et al., 2015; Bier and Burkhardt, 2019). Therefore, it is meaningful to display the variation of the nucleated contrail ice crystal number with $\Delta T := T_a - \Theta_G$ as well (see next section).

### 3.3 Final ice crystal number

In the following, we aim at summarising the results from the previous sensitivity studies. We analyse the sensitivity of final contrail ice crystal numbers to ambient temperature and relevant soot properties. Thereby, we display both the freezing fraction (left column of Fig. 5) and the absolute ice crystal number in terms of $AEI_{i,f}$ (right column of Fig. 5). The latter is shown only for sensitivities with varying $EI_s$ since otherwise there is simply a linear relationship between $\phi_{frz,f}$ and $AEI_{i,f}$.

The freezing fraction in general decreases with increasing $EI_s$ (panel a)) because the plume humidity is depleted faster at higher exhaust particle number concentrations and also distributed among more ice crystals. Hence, the smaller soot particles do not manage to form contrail ice crystals. The ensemble mean $\phi_{frz,f}$ only slightly changes with $EI_s$, which confirms the low sensitivity pointed out in Sect. 3.2.2. Accordingly, the ensemble mean $AEI_{i,f}$ increases nearly linearly with increasing $EI_s$.





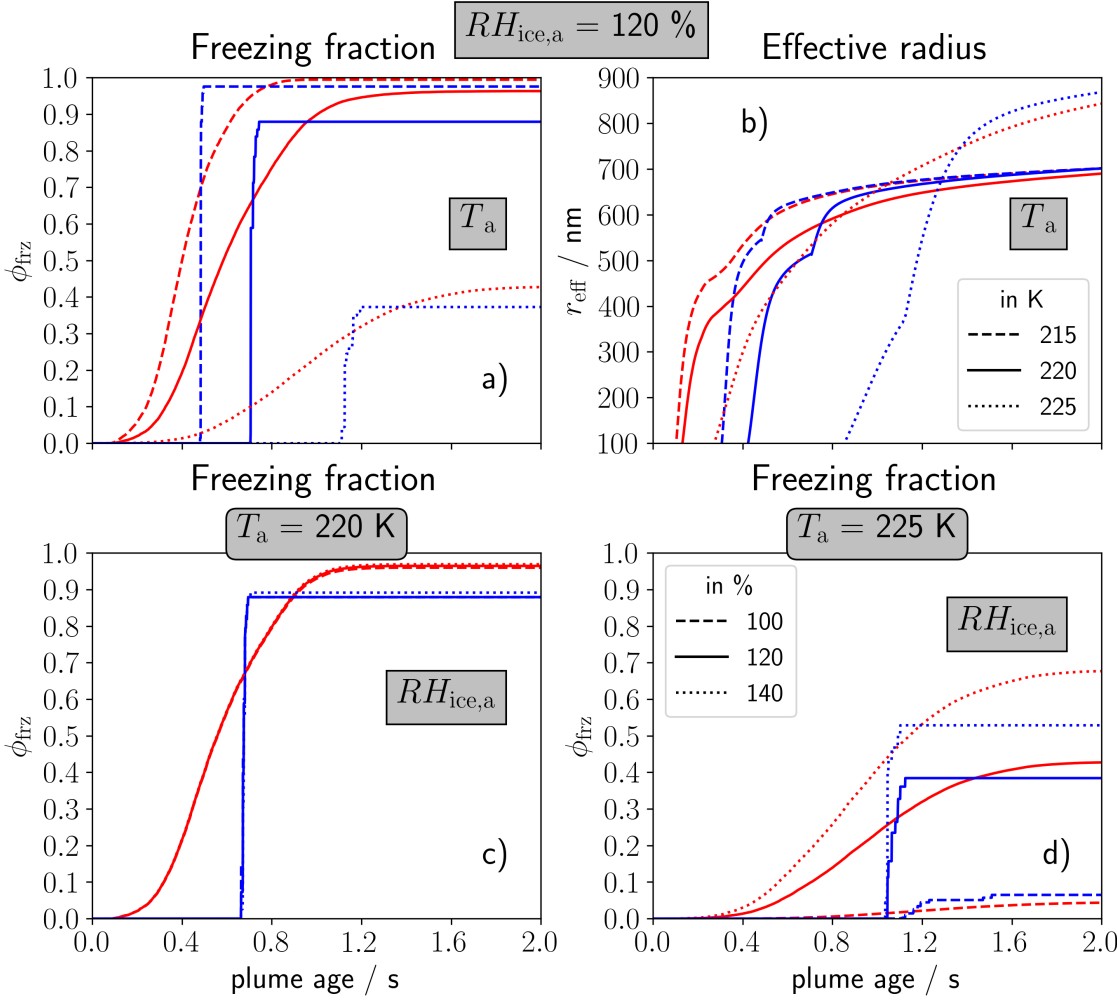

**Figure 4.** Temporal evolution of the freezing fraction of soot particles $\phi_{\mathrm{frz}}$ (panels a),c),d)) and effective radius $r_{\mathrm{eff}}$ (panel b)) for a variation of ambient temperature $T_{\mathrm{a}}$ (upper row) and $RH_{\mathrm{ice,a}}$ (lower row) according to the inserted legends in the right column. The variation of $RH_{\mathrm{ice,a}}$ is shown for two different $T_{\mathrm{a}}$ values (see panel titles).

The average traj $\phi_{\mathrm{frz,f}}$-evolution (blue curve) features a stronger decrease than the ensemble mean, in particular for the higher

soot number emissions ($EI_{\mathrm{s}} > 10^{15}\,\mathrm{kg}^{-1}$). This is due to the deactivation phenomenon explained in Sect. 3.2.2. Therefore, the increase in $AEI_{\mathrm{i,f}}$ with rising $EI_{\mathrm{s}}$ flattens in the average traj case.

Panel c) displays the variation of $\phi_{\mathrm{frz,f}}$ with ambient temperature $T_{\mathrm{a}}$ (lower label) and the difference between $T_{\mathrm{a}}$ and $\Theta_{\mathrm{G}}$ ($\Delta T$, upper label). Note that the relation between $T_{\mathrm{a}}$ and $\Delta T$ is not purely linear since the ambient relative humidity over water varies with $T_{\mathrm{a}}$ at fixed $RH_{\mathrm{ice,a}}$ and hence, $\Theta_{\mathrm{G}}$ slightly changes as well. Near the contrail formation threshold (i.e., $|\Delta T| \lesssim 3-4\,\mathrm{K}$),

$\phi_{\mathrm{frz,f}}$ strongly increases with rising $|\Delta T|$. This is because the maximum plume water-supersaturation increases and more and





more soot particles manage to activate into water droplets. Below $T_a = 220\,\mathrm{K}$ ($|\Delta T| > 6\,\mathrm{K}$), more than 90% of the soot particles turn into ice crystals so that $AEI_{i,f}$ approaches $EI_s$. The average traj simulation produces somewhat lower $\phi_{frz,f}$ values than the ensemble mean simulation. Furthermore, low soot simulations (with $EI_s$ reduced by 80% relative to the baseline case) feature a similar evolution in $\phi_{frz,f}$ and a smaller difference between the average traj and the ensemble mean (not shown).

Finally, we investigate the contrail formation dependence on the soot particle size, as specified by the geometric-mean soot dry core radius $\bar{r}_d$. Panel d) shows a variation with $\bar{r}_d$ for fixed baseline $EI_s$. At $T_a = 220\,\mathrm{K}$ (solid lines), nearly all soot particles can form ice crystals as long as $\bar{r}_d$ is larger than 20 nm. For smaller mean dry core radii, $\phi_{frz,f}$ starts to decrease. This decrease is stronger for the average traj (blue vs. red curve). Near the formation threshold (dashed), $\phi_{frz,f}$ values are clearly lower. Again, the ensemble mean values are larger than the average traj values. Moreover, we see a strong dependence on $\bar{r}_d$ over the full

parameter range.

In the third row of Fig. 5, $\bar{r}_d$ is varied together with $EI_s$ such that the overall soot mass is fixed. We find a qualitatively similar evolution in $\phi_{frz,f}$ compared to the previous sensitivity study (panel d)) so that the impact of $\bar{r}_d$ dominates. Yet, $AEI_{i,f}$ (panel f)) is controlled by the change in $EI_s$ ($EI_s$ decreases with increasing $\bar{r}_d$) and the trend is opposite to that in panel d).

The following evaluation of our box model computations relies on the comparison with other contrail formation simulations.

Generally, such model simulations can differ in various aspects, which we group into four categories: the initial and ambient conditions, the prescribed dilution, the model (micro)physics and the model framework (average traj, ensemble mean or fully 3D LES). If simulations have different settings or properties in several or all four categories, a comparison can hardly be conclusive as differences in the simulation results can have more than one origin. Hence, we made efforts to reproduce the simulation setup of the recent study by Lewellen (2020, abbr. as Lew20 hereinafter) by using his baseline conditions (more

specifically those used in his Fig. 2). They differ from our baseline conditions as summarised in our Table 1. The most significant differences are certainly the factor 10 difference in the $EI_s$-value and the shape of the soot size distribution. Lew20 either uses a monodisperse soot size distribution with $r_d = 20\,\mathrm{nm}$ or a bidisperse distribution with a 50/50 mixture of $r_d = 10\,\mathrm{nm}$ and 40 nm soot particles. Moreover, D. Lewellen was so kind to provide the dilution data that he used in his box model simulations and the simulation data depicted in his Fig. 2 [1]. This enables to perform simulations with very similar settings, basically in

all the four categories mentioned above. In particular, the microphysical treatment of contrail formation is very similar in our approach that includes the transient liquid phase.

---

[1]We received simulation data for a monodisperse soot initialisation; in the original Fig. 2 of Lew20, however results for a bidisperse soot initialisation are shown. Both simulations give basically identical results

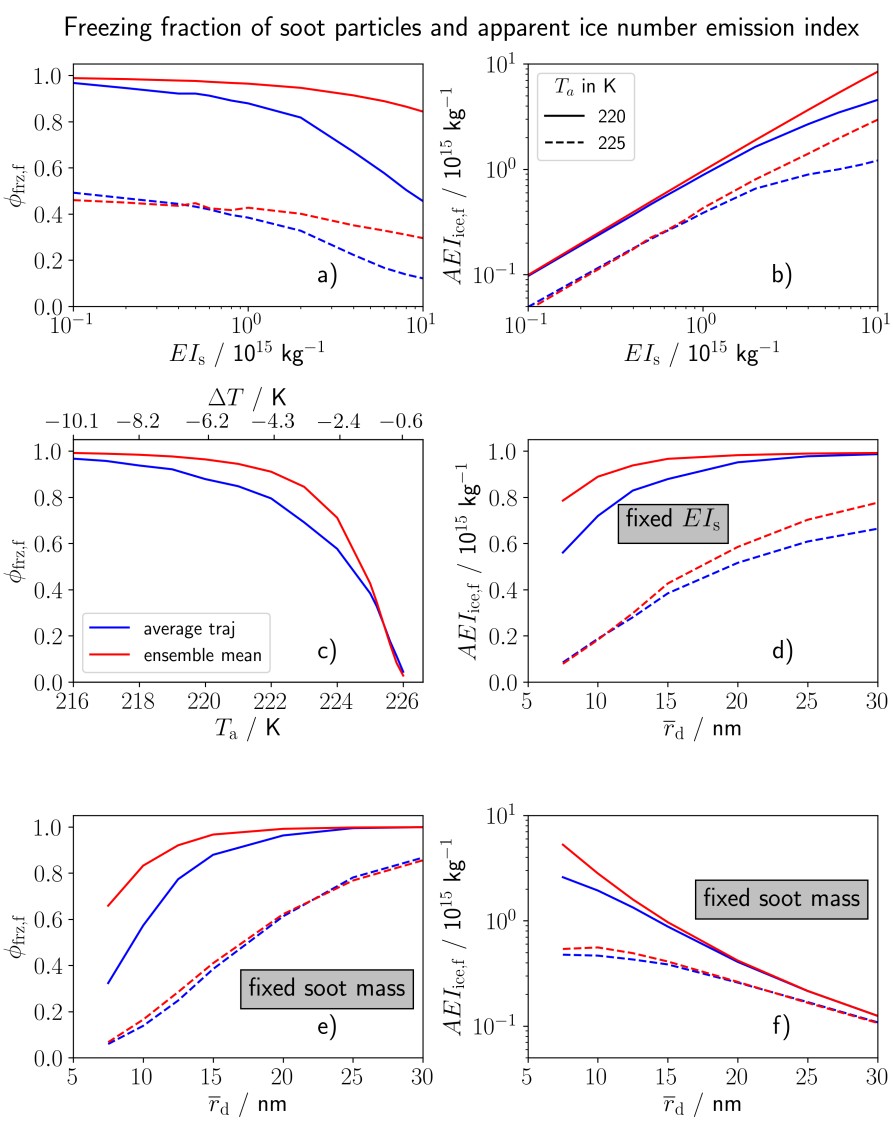

**Figure 5.** Fraction of soot particles freezing to ice crystals $\phi_{\mathrm{frz,f}}$ (left column) and apparent ice number emission index $AEI_{\mathrm{ice,f}}$ (right column) at a plume age of 2 s depending on $EI_{\mathrm{s}}$ (first row), ambient temperature $T_{\mathrm{a}}$ in panel c) (where the axis on the top additionally shows the temperature difference to the SA-threshold, $\Delta T = T_{\mathrm{a}} - \Theta_G$) and on the geometric-mean soot core radius $\bar{r}_{\mathrm{d}}$ in panels d)–f). In all but panel c) results are shown for $T_{\mathrm{a}}$ of 220 K (solid) and 225 K (dashed), indicated by the legend in the top-right panel. In panel d), $EI_{\mathrm{s}}$ is fixed to the baseline case while in panels e) and f) the total soot mass is fixed (implying an adaptation of $EI_{\mathrm{s}}$). The red lines show the ensemble mean and the blue lines the average traj results.





# 4 Evaluation

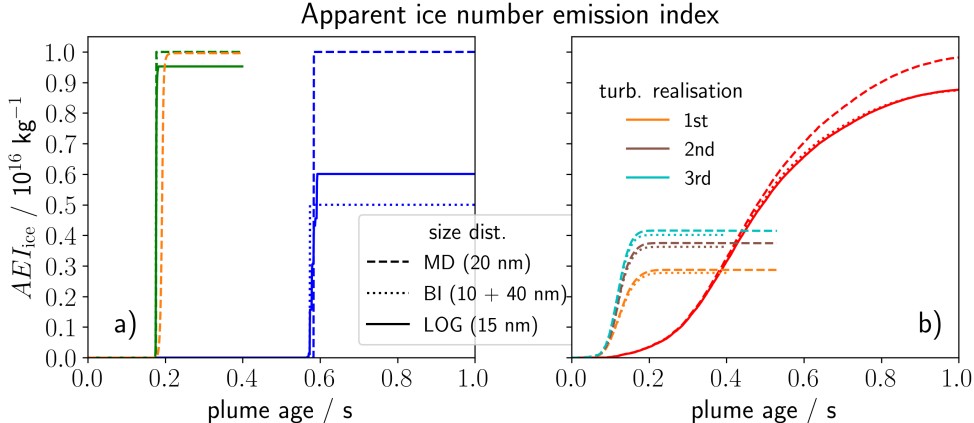

**Figure 6.** Temporal evolution of the apparent ice number emission index under baseline conditions of Lewellen (2020, "Lew20") as summarised in the bottom half of Table 1. The simulation initialisation uses a monodisperse (dashed), bidisperse (dotted) or lognormal (solid) size distribution of soot particles (see also inserted legend). The left panel juxtaposes the FLUDILES based average traj evolution (blue) with the evolution using Lew20's dilution (green) and with the results of Lew20's box model (orange). Lew20's data is available up to a plume age of 0.4 s. Note that the green dashed and dotted lines are mostly covered by the green solid line. The right panel depicts the FLUDILES ensemble mean evolution (red lines) with the LES results from Lew20's Fig. 2 for three different turbulent realisations (remaining colours).

Fig. 6 shows different types of simulations that all use the baseline conditions of Lew20. We first discuss the average traj box model simulations in the left panel. The selection of the simulations serves two purposes: First, applying two different dilution
data sets in our LCM model highlights the impact of the dilution data (green vs. blue lines). Second, the LCM simulations using Lew20's dilution data are compared to the box model simulations of Lew20 (green vs. orange lines). This aims at validating the microphysical approach and implementation in both models. Results are presented for a monodisperse and bidisperse (as in Lew20) as well as for our regular lognormal soot size distribution. All simulations feature a pulse-like increase in $AEI_{\mathrm{ice}}$ once freezing starts. However, ice crystal formation is initiated significantly earlier for Lew20's dilution, namely after 0.25 s
instead of 0.6 s for the FLUDILES dilution. As explained in Sect. 2.3, the FLUDILES dilution is too weak and the onset of contrail formation is retarded.

Using the bidisperse soot initialisation (blue dashed) we find $AEI_{\mathrm{ice}} = 0.5 EI_{\mathrm{s}}$ for the FLUDILES dilution as the 10 nm soot particles are not activated into water droplets. Yet, all soot particles are activated for Lew20's dilution. This is remarkable as for both dilution data the maximum plume water-supersaturation would be the same if the microphysical processes were
suppressed. The reason for the different activation fraction is that the time period between droplet formation and freezing is longer and correspondingly also the cooling rates are lower within this period in the FLUDILES case. This leads to a reduced peak saturation ratio, which is smaller than the critical saturation ratio for the activation of the 10 nm particles.

For the lognormal size distribution (blue solid), some smaller soot particles are only temporarily activated, but then shrink





again as the larger droplets originating from the larger soot particles deplete water vapour. With Lew20's dilution (green solid),

there is not enough time between activation and freezing to deactivate the smaller droplets. This difference originating from the different dilution data implies that the deactivation phenomenon, which we frequently encountered in our results, may not occur that frequently if plume cooling proceeds faster. Further implications concerning this aspect will be discussed in Sect. 5. As a next step, we compare our box model simulations using Lew20's dilution (green) with Lew20's box model simulation (orange). From the latter model only results for the monodisperse soot initialisation are available, yet the bidisperse case gives

basically identical results. We find a perfect agreement between both models with an only some hundredth of seconds earlier increase in $AEI_{\mathrm{ice}}$ for the LCM. (Note that the green dashed and dotted lines are mostly covered by the solid line). This is because our freezing temperature ($T_{\mathrm{frz}}$) is slightly higher than that of Lew20 (pers. communication with D. Lewellen) leading to an earlier onset of ice crystal formation.

The right panel displays FLUDILES ensemble mean simulations (red lines) and 3D LES results with online coupled mi-

crophysics of Lew20. The latter are shown for three different turbulence realisations demonstrating irreducible uncertainty involved in predicting contrail formation. The ice crystal number from the 3D LES is not only considerably reduced relative to our $AEI_{\mathrm{ice}}$ but also relative to the box model framework of Lew20 using the same microphysics as in his 3D LES. D. Lewellen explains this as follows: "Different parcels in the plume produce ice crystals at different times. As these parcels interact via mixing, the moisture consumption from early crystal growth in some parcels can prevent the activation and freezing of soot

aerosol in other parcels that later mix in. . . . Thus, the much narrower time window for ice number production seen in the box model leads to greater $EI_{\mathrm{iceno}}$ [$AEI_{\mathrm{ice}}$] than in the LES."

Originally, we supposed that switching from the average traj to the ensemble mean framework leads to more realistic quantitative results as already assumed in Paoli et al. (2008). However, we see an opposite trend compared to the 3D LES, namely the ice crystal number rises when we switch from the average traj (blue lines in left panel) to the ensemble mean (red lines in right

panel). We believe this comes from the fact that we do not consider the inter-trajectory mixing mentioned above. This means that offline ensemble simulations miss an important process and their quantitative prediction of $AEI_{\mathrm{ice}}$ is not more reliable than from an average traj simulation, even though they account for plume heterogeneity and are in several aspects more plausible. Note that the reasoning above is based on comparing only a few simulations. For instance, Lew20 shows that for $EI_{\mathrm{s}} \leq 10^{15}$, all soot particles form ice crystals in the LES (see his Figs. 5 and 11) and hence there is no longer a difference between Lew20's

average traj and LES predictions of $AEI_{\mathrm{ice}}$. Moreover, the deactivation phenomenon leading to the large difference between FLUDILES average traj and ensemble mean results may be exaggerated with the FLUDILES dilution. This implies that the difference between the average traj and the ensemble mean could be smaller with a stronger dilution. We test this hypothesis in the following discussion section.

Similar to our study, Paoli et al. (2008) compared average traj and ensemble mean results for two specific parameter settings.

They also used a set of 25000 trajectories even though it is not the same data set provided by Vancassel et al. (2014). Several aspects are similar to ours, e. g., the quasi pulse-like activation versus a formation period. Interestingly, they find that for the average traj the final ice crystal number is even three times lower than for the ensemble mean.





## 5 Discussion

In the previous section, we have shown that ice crystal formation is initiated significantly earlier and the number of formed
ice crystals may be even larger if we use Lew20's instead of the FLUDILES dilution data. Furthermore, we stressed that some
differences found between the ensemble mean and the average traj results may be particularly due to a too weak dilution. In the
following, we will estimate the time lag of the plume mixing and discuss potential improvements in our results by "accelerated"
trajectory data.

While 38% of the total combustion energy $E_{\mathrm{tot}}$ is used for the propulsion of the aircraft (according to the A340-300 propul-
sion efficiency), the remaining energy (62%) is transferred into the jet plume in the form of kinetic energy $E_{\mathrm{kin}}$ and ther-
mal energy $E_{\mathrm{therm}}$. For the initial partitioning, we assume that $E_{\mathrm{therm}}$ is in the range 51% – 31% of $E_{\mathrm{tot}}$ and, accordingly,
$E_{\mathrm{kin}} \in [11\%, 31\%] \cdot E_{\mathrm{tot}}$. During the jet expansion, $E_{\mathrm{kin}}$ is continuously converted into $E_{\mathrm{therm}}$ leading to a temperature excess
relative to a "passive tracer temperature". Based on 3D simulations with EULAG, we estimate that around 90 (95%) of $E_{\mathrm{kin}}$ is
already converted into $E_{\mathrm{therm}}$ after 0.4 s (0.8) s of plume age. Calculating the temperature excess, our estimated time lags are
0.05–0.15 s after 0.4 s, 0.1–0.25 s after 0.6 s and 0.15–0.35 s after 0.8 s of plume age. Note that the lower/upper time lag limit
refers to the smallest/largest assumed initial jet kinetic energy.

In the following, we introduce a small correction to "speed-up" our existing trajectory ensemble. More specifically: we
artificially increase the dilution rate of each FLUDILES trajectory such that in the end the average dilution of this cor-
rected trajectory ensemble, $\mathcal{D}_{\mathrm{AT,F2L}}$, matches the dilution of Lew20, $\mathcal{D}_{\mathrm{AT,Lew20}}$. At each time step, we diagnose $c_{\mathrm{F2L}}(t) :=$
$\mathcal{D}_{\mathrm{AT,Lew20}}/\mathcal{D}_{\mathrm{AT,FLUDILES}}$, where $\mathcal{D}_{\mathrm{AT,FLUDILES}}$ is the average dilution defined at the end of Sect. 2.3 and "F2L" stands for
"FLUDILES to Lewellen". For each trajectory of the existing data base, we use a modified dilution evolution defined as
$\mathcal{D}_{k,\mathrm{F2L}}(t) := c_{\mathrm{F2L}}(t) \cdot \mathcal{D}_{k,\mathrm{FLUDILES}}(t)$. It is easy to show that the average traj dilution $\mathcal{D}_{\mathrm{AT,F2L}}$ of the modified trajectory ensem-
ble equals $\mathcal{D}_{\mathrm{AT,Lew20}}$.

Fig. 7 shows the sensitivity of the final freezing fraction of soot particles ($\phi_{\mathrm{frz,f}}$) to a) $EI_{\mathrm{s}}$ and b) $T_{\mathrm{a}}$ for the original FLUDILES
simulations (blue and red lines, as already depicted in Fig. 5 a) and c)) and for the speed-up scenario (green and magenta) ac-
cording to Lew20. We are particularly interested in comparing the evolution between the ensemble mean and the average traj.
In terms of the $EI_{\mathrm{s}}$-sensitivity, the difference between the two frameworks is apparently smaller in the speed-up sceanrio than
in the original data set. Considering the variation with $T_{\mathrm{a}}$, the original average traj displays the lowest $\phi_{\mathrm{frz,f}}$ (blue) of all other
simulations since the deactivation phenomenon leads to fewer ice crystals. For the speed-up scenario, the difference between
the ensemble mean and the average traj is again significantly lower than for the original data set. Both sensitivity scenarios
confirm our hypothesis that the weak dilution of the original data is conducive to excessive deactivation in the average traj
set-up. Furthermore, the ice crystal number is in general higher for the speed-up scenario (magenta) than for the original data
set (red).

Additionally, we show the results of the parameterisation by Kärcher et al. (2015) in an adapted version. This parameterisation
was recently implemented in a global climate model (Bier and Burkhardt, 2019), where it refines the contrail initialisation



procedure. Those results (brown lines in Fig. 7) lie well in the range of the other simulations results. Overall, the extensive evaluation may help interpreting previous studies based on an average traj or a trajectory ensemble approach.

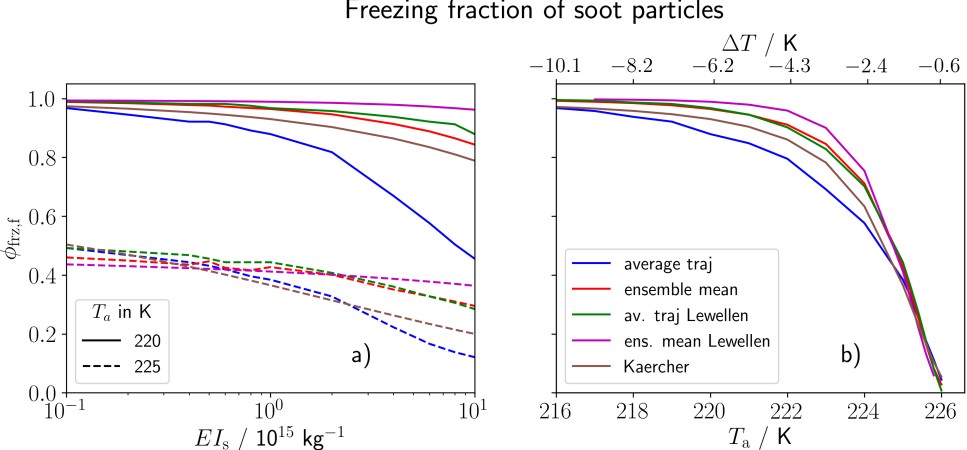

**Figure 7.** Final freezing fraction of soot particles against (a) soot number emission index and (b) ambient temperature (where the axis on top additionally shows the temperature difference to the SA-threshold $\Delta T$). We juxtapose the original FLUDILES trajectory ensemble (red) with the speed-up ensemble (magenta), whose average dilution matches that of Lewellen (2020). The blue and green lines show the corresponding average traj evolution of both ensembles. Additionally, the results from the parameterisation of Kärcher et al. (2015) with an adaption regarding the calculation of the activation dry core radius of soot particles (brown) are shown.

## 6   Conclusions and outlook

In this paper, the particle-based Lagrangian Cloud Module (LCM) (Sölch and Kärcher, 2010) has been extended to cover the
specific microphysics of contrail formation. One of our objectives was to test the newly extended LCM model in a simple box model framework prior to its coupling to the 3D LES model EULAG. Furthermore, our aim was to analyse the most relevant sensitivities of contrail ice crystal numbers to various soot properties and ambient conditions and to compare our results with previous studies.

We employ a data set with dilution histories along 25000 trajectories sampling the expanding jet. These data are provided by
3D FLUDILES simulations behind the engine nozzle of an A340-300 aircraft (Vancassel et al., 2014). We compare a single box model run with an average dilution with an ensemble mean of box model runs for all dilution histories.

Plume mixing leads to a strong cooling at the plume edge where water-saturation is reached first. Later on, also the plume centre cools. Therefore, soot particles form water droplets (and subsequently freeze into ice crystals) first near the plume edge and afterwards in the plume centre.

Consistent with previous studies (e.g., Kärcher and Yu, 2009; Kärcher et al., 2015; Bier and Burkhardt, 2019), contrail ice crystal number varies strongly with atmospheric parameters (like ambient temperature and relative humidity) only near the





contrail formation threshold. Thereby, the difference between the ambient and Schmidt-Appleman threshold temperature is of relevance since it controls the maximum water-supersaturation in the plume.

Investigating the impact of soot properties on contrail evolution, we find that the fraction of soot particles freezing to ice crystals significantly changes with mean size and solubility of the soot particles only near the contrail formation threshold. This is due to the combined effect of the non-linear Kelvin term and the lognormal soot particle size distribution. Absolute ice crystal numbers are, on the other hand, controlled by the soot number emission index for all ambient conditions. To investigate the impact of current biofuel blends, we analysed a combined variation of the soot number emission index and the average soot core radius. We find that the decrease in contrail ice crystal number does not only result from the reduced soot particle number but also from the smaller soot particle sizes.

We evaluate our study with results from a recent study of Lewellen (2020) who uses a similar contrail formation microphysics with a transient liquid phase. Prescribing the same initial and ambient conditions, we particularly find a later onset of contrail formation in our model. This is because our dilution is too weak as discussed in the previous section. When employing the dilution data provided by Lewellen (2020), we find a perfect agreement in the evolution of contrail ice crystal number suggesting correct implementations of the microphysics in both models.

Switching from a single trajectory to the ensemble mean framework shows only an improvement in terms of a more continuously (instead of pulse-like) evolution in contrail ice nucleation. On the other hand, we see an increase in final ice crystal numbers which is, at least for high soot number emissions, an opposite trend compared to the recent 3D LES results of Lewellen (2020). This is likely due to the fact that we do not consider the interaction (mixing and depletion of water vapour) between the different trajectories and, therefore, overestimate ice crystal formation. Hence, using a large spatially resolving trajectory ensemble does not necessarily lead to improved scientific results contrary to what we expected in the beginning.

The presented aerosol and microphysics scheme describing contrail formation is of intermediate complexity and thus suited to be incorporated within 3D simulations (LES) of contrail formation explicitly simulating the jet expansion. Or next task is to extend the EULAG-LCM model system, which has been extensively used for high-resolution contrail simulations during the vortex and dispersion phase, by the specific contrail formation physics.

*Data availability.* The presented data are available from the corresponding author upon request (Andreas.Bier@dlr.de).





## Appendix A: Equations for water and ice microphysics

### A1 Condensational growth

The binary diffusion coefficient of air and water vapour $D_\mathrm{v}$ is calculated as (Pruppacher and Klett, 1997)

$$D_\mathrm{v}/\left(\mathrm{m}^2\,\mathrm{s}^{-1}\right) = 2.11\cdot 10^{-5}\left(\frac{p_0}{p_\mathrm{a}}\right)\left(\frac{T}{T_\mathrm{mlt}}\right)^{1.94} \tag{A1}$$

with $p_0 = 1013.25\,\mathrm{hPa}$ denoting the standard surface pressure and $T_\mathrm{mlt} = 273.15\,\mathrm{K}$ the equilibrium freezing temperature over pure ice water. The equation is actually valid for $233.15\,\mathrm{K} < T < 313.15\,\mathrm{K}$ but we use it for lower temperatures as well. The heat conductivity of air is approximated as a function of temperature (Jacobson, 1999)

$$\hat{K}/\left(\mathrm{J}\,\mathrm{m}^{-1}\,\mathrm{s}^{-1}\,\mathrm{K}^{-1}\right) = 2.381\cdot 10^{-2} + 7.113\cdot 10^{-5}\cdot(T - T_\mathrm{mlt})\,\mathrm{K}^{-1}, \tag{A2}$$

and the specific latent heat for condensation/evaporation $L_\mathrm{c}$ is obtained from the equation of Rogers and Yau (1996)

$$L_\mathrm{c}/\left(\mathrm{J}\,\mathrm{kg}^{-1}\right) = 2.5\cdot 10^6 - 2.36\cdot 10^3\cdot(T - T_\mathrm{mlt})\,\mathrm{K}^{-1} + 1.6\cdot(T - T_\mathrm{mlt})^2\,\mathrm{K}^{-2} - 0.06\cdot(T - T_\mathrm{mlt})^3\,\mathrm{K}^{-3}. \tag{A3}$$

If droplet radii are initially small, i. e., having a magnitude comparable with the molecular mean free path, the condensational growth in the transition regime has to be considered. We use the correction terms for the mass and heat term in Eq. (2) according to Fuchs and Sutugin (1971):

$$\beta_\mathrm{m} = \frac{Kn_\mathrm{v} + 1}{(b_1/\alpha_{m,c} + b_2)\cdot Kn_\mathrm{v} + b_1/\alpha_{m,c}\cdot Kn_\mathrm{v}^2}, \tag{A4}$$

$$\beta_\mathrm{t} = \frac{Kn_\mathrm{a} + 1}{(b_1/\alpha_{t,c} + b_2)\cdot Kn_\mathrm{a} + b_1/\alpha_{t,c}\cdot Kn_\mathrm{a}^2}, \tag{A5}$$

where $b_1 = 4/3$ and $b_2 = 0.377$ are empirical constants. The Knudsen number ($Kn$) describes the ratio between the mean free path of the surrounding gas molecules and the droplet radius so that

$$Kn_\mathrm{v} = \lambda_\mathrm{v}\,r^{-1}, \tag{A6}$$

$$Kn_\mathrm{a} = \lambda_\mathrm{a}\,r^{-1}. \tag{A7}$$

The mean free paths of vapour ($\lambda_\mathrm{v}$) and air molecules ($\lambda_\mathrm{a}$) are given by Williams and Loyalka (1991)

$$\lambda_\mathrm{v} = 2\cdot D_\mathrm{v}\sqrt{\frac{1}{2\,R_\mathrm{v}\,T}}, \tag{A8}$$

$$\lambda_\mathrm{a} = 0.8\cdot\frac{\hat{K}\,T}{p_\mathrm{a}}\sqrt{\frac{1}{2\,R_\mathrm{d}\,T}}, \tag{A9}$$

with $R_\mathrm{d}$ and $R_\mathrm{v}$ denoting the specific gas constants for dry air and vapour, respectively.
$\alpha_\mathrm{m,c}$ and $\alpha_\mathrm{t,c}$ are the mass and thermal accommodation coefficients for condensation. Here, we set both parameters to unity since droplet growth rates from cloud model studies agree best with experimental studies when choosing these values (Laaksonen et al., 2005).





## A2 Depositional growth

The specific latent heat for deposition/sublimation is approximated according to Rogers and Yau (1996)

$$L_{\mathrm{d}}/\left(\mathrm{J\,kg}^{-1}\right) = 2.8346 \cdot 10^{6} - 340 \cdot (T - T_{\mathrm{mlt}})\,\mathrm{K}^{-1} - 10.46 \cdot (T - T_{\mathrm{mlt}})^{2}\,\mathrm{K}^{-2}. \tag{A10}$$

The transitional correction factors that appear in Eq. (7) are given by Pruppacher and Klett (1997)

$$\beta_{\mathrm{v}} = \frac{r}{r + \lambda_{\mathrm{v}}} + \frac{4\,C\,D_{\mathrm{v}}}{\alpha_{\mathrm{m,d}}\,v_{\mathrm{th}}\,r}, \tag{A11}$$

$$\beta_{k} = \frac{r}{r + \lambda_{\mathrm{a}}} + \frac{4\hat{K}}{\alpha_{\mathrm{t,d}}\,v_{\mathrm{th}}\,c_{\mathrm{p}}\,\rho_{\mathrm{a}}}, \tag{A12}$$

where $\rho_{\mathrm{a}}$ and $c_{\mathrm{p}}$ are mass density and specific heat constant of dry air and $\alpha_{\mathrm{m,d}}$ and $\alpha_{\mathrm{t,d}}$ represent the mass and thermal accommodation coefficients for deposition, respectively. In this study, we set $\alpha_{\mathrm{m,d}}$ to 0.5 according to Kärcher (2003) and $\alpha_{\mathrm{t,d}}$ to 0.7.

## Appendix B: Determination of the Koehler curve maximum

As introduced in Sect. 2.2.3, the Kappa-Köhler equation is given by (Petters and Kreidenweis, 2007)

$$S_{\mathrm{K}} = \frac{r^{3} - r_{\mathrm{d}}{}^{3}}{r^{3} - r_{\mathrm{d}}{}^{3}(1 - \kappa)} \cdot \exp\left(\frac{2\,\sigma\,M_{\mathrm{w}}}{R\,T\,\rho_{\mathrm{w}}\,r}\right). \tag{B1}$$

To obtain the critical saturation ratio for activation of soot particles into water droplets ($S_{\mathrm{c}}$), we search for the local maximum of $S_{\mathrm{K}}$ which requires $\frac{\partial S_{\mathrm{K}}}{\partial r}$ to be zero. The root $r_{\mathrm{c}}$ is called the critical radius for activation. Substituting $\alpha = 1 - \kappa$ and $Ke = \frac{2\,\sigma\,M_{\mathrm{w}}}{R\,T\,\rho_{\mathrm{w}}}$ yields

$$\left.\frac{\partial S_{\mathrm{K}}}{\partial r}\right|_{r_{\mathrm{c}}} = \left(\frac{-Ke\,r_{\mathrm{c}}{}^{6} - 3\,r_{\mathrm{d}}{}^{3}\,(\alpha - 1)\,r_{\mathrm{c}}{}^{4} + r_{\mathrm{d}}{}^{3}\,r_{\mathrm{c}}{}^{3}\,Ke\,(\alpha + 1) - \alpha\,Ke\,r_{\mathrm{d}}{}^{6}}{(r_{\mathrm{c}}{}^{4} - r_{\mathrm{d}}{}^{3}\,\alpha\,r_{\mathrm{c}})^{2}}\right) \cdot \exp\left(\frac{Ke}{r_{\mathrm{c}}}\right) = 0. \tag{B2}$$

The determination of $r_{\mathrm{c}}$ requires the nominator of the first term

$$f(r_{\mathrm{c}}) = -Ke\,r_{\mathrm{c}}{}^{6} - 3\,r_{\mathrm{d}}{}^{3}\,(\alpha - 1)\,r_{\mathrm{c}}{}^{4} + r_{\mathrm{d}}{}^{3}\,Ke\,(\alpha + 1) - \alpha\,Ke\,r_{\mathrm{d}}{}^{6} \tag{B3}$$

to be zero. We solve this numerically and apply a Newtonian iteration until the root $r_{\mathrm{c}}$ is determined with an accuracy of $< 0.01\,\mathrm{nm}$.

Figure B1 a) shows that $S_{\mathrm{c}}$ increases with decreasing $r_{\mathrm{d}}$ due to the Kelvin effect. The variation of $S_{\mathrm{c}}$ with temperature $T$ is quite low for the range where activation in exhaust plumes typically occurs (see legend). Using the simplified Eq. (10) of Petters and Kreidenweis (2007) (green line) causes a significant overestimation of $S_{\mathrm{c}}$ for $r_{d} < 15\,\mathrm{nm}$. The critical saturation ratio increases with decreasing hygroscopicity parameter $\kappa$ due to the solution effect (Fig. B1 b)).

©c Author(s) 2021. CC BY 4.0 License.





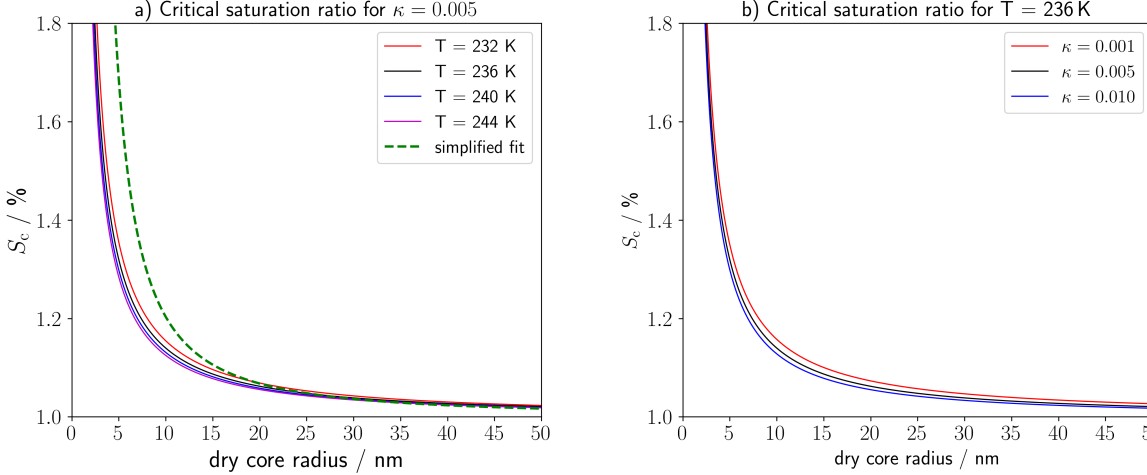

**Figure B1.** Critical saturation ratio ($S_c$) for the activation of soot particles against dry core radius for different (a) plume temperatures and (b) hygroscopicity parameter. The dashed line in (a) depicts the evolution of $S_c$ for the simplified Eq. (10) of Petters and Kreidenweis (2007) which is valid for $\kappa >= 0.1$.

*Author contributions.* Conceptualisation: A. Bier and S. Unterstrasser; Methodology: A. Bier and S. Unterstrasser; Software Programming: A. Bier; Validation: S. Unterstrasser; Formal Analysis: A. Bier and S. Unterstrasser; Investigation: A. Bier; Resources: X. Vancassel; Original
draft: A. Bier; Writing—Review and Editing: all authors; Visualisation: A. Bier; Funding Acquisition: A. Bier.

*Competing interests.* There are no competing interests.

*Acknowledgements.* The scientific work has been funded by the Deutsche Forschungsgemeinschaft (DFG) within the project "BI 2128/1-1". We thank David Lewellen for providing us with simulation and dilution data published in Lewellen (2020) and we thank Klaus Gierens for his internal review.





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
