# Peer review of "Box model trajectory studies of contrail formation using a particle-based cloud microphysics scheme"

_Atmospheric Chemistry and Physics, 2021_

## Author Comment (AC1)

Dear David,

we very much appreciate your important and constructive hints and comments. Please find our answers below (the reviewer's comments are repeated in italics).

*This is primarily a ``model development" paper, extending an existing microphysical cloud model (LCM) to allow simulation of contrail formation. This is implemented at a box-model level as a step to future incorporation within a 3D LES. Most of the text is devoted to describing the necessary additions to the microphysics and to testing the resulting model with a small parametric box-model study. Neither covers particularly new ground in the literature, the added microphysics being pieces already utilized for this purpose by prior researchers, and the box-model parametric study results being consistent with, and much less extensive than, prior studies using box models (e.g., Kärcher and Yu 2009) over even full LES (Lewellen 2020). Nonetheless, given the complexity of such simulation codes, the successful comparison presented provides a useful crosscheck on the prior work, and the generally clear documentation will provide a useful reference as the authors proceed with future model development and contrail studies. The most novel part of the present work are the simulations run in the ``ensemble" mode, in which box-model simulations using dilution histories from a large representative sampling of Lagrangian trajectories from a 3D LES are averaged over. The ultimate utility of this method is, however, highly questionable (for reasons discussed below) and so the main benefit here may be as a cautionary tale. Perhaps not surprisingly, my review here is weighted more heavily to the comparisons in the text with my own work (Lewellen (2020), denoted Lew20 hereafter).*

*Main points:*

*(1) The paper makes clear that the eventual goal is to incorporate the modified LCM within a particular LES (EULAG) for 3D contrail formation studies. But it doesn't motivate why the use of this computationally much more expensive approach may be required, or give any hints of the hurdles to be overcome for using EULAG for this purpose. A brief statement or referencing for both would seem useful to the reader. Lew20 provides relevant content on both counts: the simulations there illustrate that in some broad regions of the physical parameter space the box model results differ substantially from the LES ones, and several of the difficulties involved in modifying an LES (even one already capable of simulating aging contrails) for the distinct environment required to study initial contrail formation are described.*

Before the formulation of our eventual goal, we made clear that there are (except of your recent study) basically two complementary model approaches to investigate contrail formation which either focus on jet dynamics or on plume microphysics. Both approaches suffer from certain strengths and weaknesses and our basic goal is to combine these two "worlds". However, the disadvantages of the recent 0D box model studies were not properly described such that the motivation of the prospective 3D studies might not have been clear enough. Therefore, we have added in section 1:

"However, they neglect the large spatial variability in jet plumes arising from turbulent mixing of the hot exhaust with ambient air and simulate only an average mixing state over the whole plume cross sectional area. Spatial inhomogeneities (in particular, strong radial gradients and turbulent perturbations) in plume temperature and relative humidity are not considered. This leads to uncertainties in representing contrail ice crystal formation wit unclear implications on contrail properties during the subsequent vortex and dispersion phase."

Moreover, we added the following lines to the manuscript:

"Many findings related to contrail formation rely on results from box models or analytical approaches with a crude representation of plume heterogeneity. By employing the same microphysics in 3D LES and in a box model with a mean dilution derived from the former model, Lewellen (2020) highlights for which parameter configurations findings drawn from box model results are similar to those from 3D LES. Contrail ice crystal numbers per burnt fuel mass are consistent in both model frameworks for low exhaust particle numbers or in scenarios where ice crystals mainly form on ambient aerosol particles. However, the box model generally overestimates these ice crystal numbers for soot particle numbers of current engines/fuels. Moreover, the ice crystal formation on emitted ultrafine volatile particles becomes more substantial in the LES than in the box model."

One hurdle to use EULAG for our purpose is that past studies with the model system EULAG-LCM have been performed with an EULAG version that does not support the compressible and gas dynamics equations. Therefore, the LCM model needs to be coupled to an up-to-date EULAG version that accounts for compressibility effects in the jet plume. This aspect is now also mentioned in the manuscript.

*(2) I think that variable-density/compressibility effects are not being properly taken into account in the plume dilution equations in sections 2.3.1 and 2.3.2 where it is assumed that temperature dilutes like a passive tracer. Masses are conserved in the exhaust plume, but density and temperature are (by the ideal gas law and the near constancy of pressure in a free jet) inversely related. For temperature to be conserved as a passive tracer to good approximation then requires the temperature differences encountered to be small relative to the absolute temperature. While this is a reasonable approximation in most atmospheric models, it does not generally hold in the hot jet exhaust plume until sufficient dilution takes place. The nature and limits of the approximation being made here deserves at least a mention in the text. This deviation of T from a passive tracer is in addition to that from the conversion of jet kinetic energy to thermal energy (noted in line 270). Treating T as evolving like a passive tracer with no corrections (as assumed) will lead to the largest inaccuracies here for the ``ensemble'' cases because contrail formation commences earlier there and because that relation has been used to define the ensemble itself via equation (17).*

Assume we mix two air parcels (indices 1 and 2 for the two separate parcels) and mix them (index "mix" for the mixed parcel), then the conservation of tracer mass can be expressed in terms of tracer mixing ratios $\chi$ as follows:

$\chi_1 * \rho_1 * V_1 + \chi_2 * \rho_2 * V_2 = \chi_{mix} * \rho_{mix} * V_{mix}$.

An analogous equation for T follows from the conservation of internal energy (in isobaric processes):

$T_1 * \rho_1 * V_1 + T_2 * \rho_2 * V_2 = T_{mix} * \rho_{mix} * V_{mix}$.

$\rho$ is the density of air and V is the parcel volume.

Note that both equations have the same functional form. From the first and second equation, Eqs. (8) and (10) in the manuscript can be derived, respectively. Eq. (10) follows from conservation of internal energy and we believe the validity of this equation is not restricted.

However, we agree that there is a degree of freedom in defining the average trajectory and the ensemble mean quantities. For the derivation of the average trajectory, one can require that the temperature, the cross-section or the mass of the average trajectory is identical to the average temperature or the total cross-section or total mass of the trajectory ensemble, respectively. We opted for requiring that the temperatures match. However, we conclude after the revision period that a mass-weighted averaging would have been more appropriate since the internal energy of the average traj and

traj ensemble were then identical. Consistent with that, the ensemble mean quantities should be mass-weighted quantities. However, we employed an area-weighting in our original presentation. But for most quantities that we displayed the weighting cancels out. Hence the computations and evolutions of $\phi_{act}$, $\phi_{frz}$ and $r_{eff}$ do not change, when we switch from an area- to a mass-weighting. The only displayed ensemble quantity that is affected from the switch is the ensemble mean relative humidity. Therefore, we updated Fig 2a/b accordingly yielding marginal differences with slightly lower peaks of the "new" RH values.

Moreover, we added explanations at the end of section 2.3.2 and section 2.4.2 to remove any ambiguities, respectively:

"There are other ways of computing the mean properties of the trajectory ensemble. E.g. the mean temperature can be a weighted average, where the trajectories' cross-sections or masses are used as weights. The internal energy is conserved when a mass-weighted temperature average is used. The Supplement shows additional results for a mass-weighted temperature average and contrasts them with our default procedure."

and

"For ensemble mean values, intensive and extensive physical quantities are mass-weighted averages and sums of trajectory ensemble values, respectively. For quantities that are defined as ratios like the effective radius or relative humidity, first the averages of the quantities that appear in the denominator and nominator are computed and then the ratio over those averaged quantities is taken. Note that the mass weights cancel out in the definitions of $\phi_{act/frz}$ and $r_{eff}$."

In a new sensitivity test, we use a mass-based weighting consistently for both, the derivation of the average traj and the mean quantities of the ensemble simulations. The ensemble mean results are only marginally affected as explained above. Hence, we focus on the differences between prescribing an **unweighted** and a **mass-weighted average temperature** in the "average traj" framework and present the analysis in a newly added supplement:

For the mass-based weighting, the mean ("passive tracer") temperature decreases faster than without weighting so that the dilution becomes on average stronger. Our sensitivity test shows that water saturation in the plume and the onset of droplet formation occur earlier (by around $0.05 - 0.15$ s for the considered cases). The final freezing fraction of soot particles is slightly higher (by few pp) for the mass-weighted than for the unweighted average traj.

[Unfortunately, a consistent treatment throughout the whole study would require to repeat all "average traj" simulations using a different temperature/dilution history. Given the fact, that the missing inter-trajectory mixing is the biggest shortcoming (besides temperature not being a real passive tracer in FLUDILES) that leads to differences between the "average traj" and the "ensemble mean" frameworks, we refrain from redoing all "average traj" simulations.]

*(3) A sizable portion of the paper is spent on the ``ensemble'' trajectory approach. Even while using a large number of trajectories from an LES run, thus providing some detailed spatial information, this approach leaves out a crucial piece of physics: the microphysics along each of the trajectories is computed independently of what is happening on the others. In reality all these various parcels are mixing with each other so that moisture condensed on aerosol in one parcel will not be available later to condense on aerosol in other parcels it mixes with. It is this competition between different parcels that Lew20 concluded was responsible for box-model results sometimes greatly over-predicting ice number relative to LES results. The ``ensemble'' mode gets this entirely wrong, predicting a higher ice number relative to the ``average'' box model results. These shortcomings are noted eventually in the main body of the paper (pg. 24), but only after promising statements are made about the method earlier*

*(e.g., line 88), and they are not reflected in the abstract or conclusions. This likely gives a misleading impression of the potential utility of the method. I would suggest remedying this omission. Indeed, the illustration of this cautionary tale could be the most useful new result in the present paper, given that others may be tempted to use this approach in different applications (or have already). It would also be worth explaining (if the authors know) why the ``ensemble'' freezing fraction always lies above that for the ``average'' box model when they differ. I suspect this may involve the specific averaging procedures employed, which are not specified in detail in the text and seem to rely on treating temperature as a passive tracer (c.f. lines 286-287).*

We consider it one of the main messages of the manuscript that the ensemble mean approach with independent individual trajectories does not lead to better results than a single trajectory approach. We believe that earlier papers that used such a multi-trajectory approach have not worked out this so clearly.

Based on your comments, however, it seems that we have to point out our finding on the deficiencies of the multi-trajectory approach more clearly. Note that this issue is already discussed in the evaluation section 4. Corresponding remarks are also given in the next-to-last paragraph of the conclusions, but statements in the abstract were missing. Therefore, we have added

"Using an ensemble mean framework instead of a single trajectory does not necessarily lead to an improved scientific outcome. Contrail ice crystal numbers tend to be overestimated since the interaction between the different trajectories is not considered." at the end of the abstract.

In fact, the last paragraph of section 2.3.2 is unspecific about our averaging procedure. Our answer to your previous comment already covered this issue.

*(4) Some of the statements in comparison to Lew20 are perhaps misleading. First it is stated (lines 20, 614) that the results of the present model and Lew20 on a comparison case are in excellent agreement. This is true when comparing to the box model results of Lew20 (which does provide a useful check on the microphysics) but not with the LES simulation results from Lew20 of the same case, which differ significantly (i.e., compared to the best results of Lew20 on this case the agreement is in fact not so good). Also, referring to this as a cross-validation (line 20) is too strong: since the present work uses many of the same parameterizations in the microphysics as Lew20, it would be possible to have excellent agreement between the two models even if one of these parameterizations were to turn out to be physically poor.*

This is an important hint. The comparison mentioned in the abstract refers to the box model results of Lew20. We now explicitly write "Comparing **with box model results** of a recent contrail formation study by Lewellen (2020) (using similar microphysics) shows […]"

The word validation was not properly used. Typically, a validation exercise implies a comparison that can tell us if the implemented physics is appropriate. What we wanted to express should be referred to as verification. Both models use similar physics and a favourable agreement between the two makes it likely that correct solution procedures are implemented. Hence, we replaced "cross-validating" by "cross-verifying" in the abstract.

*Later, it is suggested (e.g., lines 64-67, 78-79) that the scheme of representation of particle sizes in the present model will improve upon that utilized in Lew20, but that is not at all clear. Although Lew20's LES model itself will allow for more complicated soot spectra (at increased numerical cost), Lew20 presented results only for mono-disperse and bi-disperse soot spectra. This was sufficient to show through examples that the final results were not very sensitive to the details of this size initialization, justifying the simplification, and directly contradicting the supposition here in line 65-67 that it might*

*lead to too narrow ice spectra.*

Since we prescribe log-normally distributed soot particle spectra (with radii ranging from a few to hundreds of nm) that are consistent with laboratory measurements (e.g., Petzold et al. 1999)), we have an improved soot initialisation compared to the mono- or bidisperse distribution in Lewellen (2020). Anyway, we have now formulated our statement more neutrally:

"The microphysical parameterisation is similar to that in Lewellen (2020) but the numerical approach of the microphysics differs as our study relies on a particle-based description and not on a Eulerian spectral bin model. Moreover, we prescribe soot particles by a lognormal instead of a monodisperse or bidisperse size distribution."

We find that it is not sufficient to conclude only from monodisperse and bidisperse soot spectrum initialisations that the final results would be not very sensitive to the details of the exhaust particle size initialisation. In our manuscript, we varied a broad range of different soot properties including variations in the size distribution. Thereby, we see for instance a high sensitivity of the final contrail ice crystal number to the average soot dry core size (Fig. 5d) over the whole considered parameter range at near-threshold conditions and for small dry core radii even at lower ambient temperature. Furthermore, Fig. 6a (blue lines) displays a clear difference in the evolution of contrail ice crystal number between the log-normal and the bimodal/monodisperse distribution even though that difference becomes lower if the dilution is stronger (green lines).

*Lew20 found in practice that having well-resolved droplet and ice size-spectra was the much more critical requirement to properly representing competition between different sized aerosol populations in the exhaust plume. In that regard the binned microphysics representation of Lew20 would seem to have the advantage over the particle-based description used here. Given the modest number of SIPs utilized here to represent the entire spectral shape (c.f., lines 143-146), the binned scheme with a healthy number of bins (as in Lew20) can more faithfully represent size spectra, particularly in the tails of the distributions that can play an out-sized role.*

We do not agree with your statement that bin models are better suited than particle-based approaches. In the cloud physics community (with a focus on natural liquid clouds) particle-based approaches are expected to overcome several long-lasting shortcomings/deficiencies of bin microphysics (numerical dispersion in mass space, curse of dimensionality etc.). Of course, not all of the advantages raised in Grabowski et al. (2019) are relevant for the current simulation problem. But the two mentioned issues are also relevant in your simulations: the term "curse of dimensionality" describes the fact that multi-dimensional binned attribute spaces are basically not feasible (you encounter this as well as it makes it unfavourable to initialise a continuous aerosol distribution in your bin model). Moreover, numerical dispersion effects in diffusional growth processes might affect your simulation results. Indeed, it would be interesting to see whether the different numerics (Lagrangian/Eulerian microphysics) play a critical role in contrail formation simulations and more generally in contrail simulations of all phases.

Unterstrasser & Sölch (2014) presented convergences test and demonstrated that relatively few simulation particles (SIP) suffice for physically converging results in contrail-cirrus simulations. Clearly, such convergence tests have to be performed for any new type of simulation. And indeed, such tests have been performed for the current study. Moreover, our SIP initialisation technique is flexible enough to create a "*nice*" initial SIP ensemble. Unterstrasser et al. (2017c) define the properties of such "nice" SIP ensembles and showed the importance of the SIP initialisation procedure for the performance of particle-based collisional growth algorithms. This shows that we critically questioned

the performance of our LCM in the past and devised the current setup with the same scrutiny.

*(5) The explanation of the differences between the results for the dilution histories from FLUDILES and Lew20 (lines 509 and following and section 5) could be sharpened significantly by making use of additional results given in Lew20. The authors attribute the slower plume dilution in the FLUDILES case to using temperature to deduce the dilution history without properly accounting for the conversion of jet kinetic energy to heat. This is a likely contributing factor (as perhaps is the issue in item (2) above), but a large part of the dilution rate difference between the FLUDILES and the baseline Lew20 case seems to be a perfectly physical one: the former simulation is apparently for a larger engine than the latter (initial plume radius of 0.5 m as stated in line 255 for the former, versus an initial plume core radius of 0.3 m in the Lew20 base case). Lew20 includes an analysis of the effects of increasing engine size, including the resulting increase in dilution time scale. Further increasing the dilution time scale is the choice in FLUDILES of an excessively smeared initial profile at the plume edge (line 264-265 and fig. 1a). Finally, Lew20 included a large parametric study varying the engine size (and hence dilution rate) and explained the differences in ice numbers that resulted. The variations with dilution-rate seen here in fig.7 all seem to conform (at least qualitatively) with those prior results.*

Thank you for pointing out that the engine size and the associated size of the nozzle and initial plume diameter play a role in the dilution speed. Therefore, we included

"Another aspect of physical nature is our higher initial plume radius (i.e., 0.5 m instead of 0.3 m in Lew20) causing the exhaust plume to dilute on a longer time scale. Lew20 analysed the effects of increasing engine size and found significant impacts on contrail ice crystal formation particularly for higher aerosol number emission indices (see his Figs. 13 and 14)." in the evaluation section.

Clearly, FLUDILES used an artificially smoothed initial profile, which is demonstrated in Fig 1a. We do not fully share your statement that this smoothing is excessive. Having in mind that the plume radius $r_P$ increases by a factor of nearly 20 within 1s, we believe that the initial smoothing creates an offset in plume age of a few hundredths of a second.

*(6) While the simulation cases are generally well described, some of the specifications are missing, vague, or departing from physical expectations. What engines are assumed? Is the bypass treated and if so, how? The radial gradient of the plume at the engine exit (c.f., fig. 1a) seems unphysically spread out. Line 265 implies this is for numerical reasons (why?). And given this gradient, how is r_p (line 343) actually evaluated? Lines 297-298 state that the fuel consumption has been adjusted, but the description why is vague and the value of m_f is never actually given. Under normal cruise conditions engine parameters (and hence properties of the exhaust jet) will generally shift some with ambient temperature. Are these effects at all included in the T_a varying cases considered? If not, that should be noted. Some of these questions may be addressed in Vancassel et al. (2014), but the present paper should be self-contained on the basic specifications and levels of approximations employed.*

The FLUDILES data are based on the **CFM56** engine family and the separation into a core and bypass flow was not considered.

It is a typical procedure in Eulerian (grid-based) models that initial prognostic fields should not feature step function-like discontinuities and hence such steps are smoothed. We have added

"This ensures numerical stability as too strong gradients in prognostic variables are numerically intricate." after introducing our initial temperature profile in section 2.3.2. We wonder if you do not have to include such precautionary measures in your model, but this depends on the numerical scheme that solves the advective terms.

In our opinion, this smoothing procedure is not necessarily unphysical since we prescribe the water

vapour consistently with temperature / dilution for each trajectory. Moreover, the initial profile should not be relevant for our study as radial distributions should have lost the information of the initialisation (see section 5 on self-similar jet flows in Pope (2000)) and the contrail microphysics occurs when the plume is diluted.

The plume radius $r_p$ is evaluated as the distance of the most remote trajectory from the plume centre at a given plume age and it is displayed as black line in Fig 1b.

The determination/adaption of the fuel consumption is now explained more precisely. The slight shift of engine/fuel parameters with varying ambient conditions is not considered in the manuscript. We do not think it would be a good idea to intermingle different effects into our basic parametric studies. These issues are now described in section 2.4.1.

*(7) The physical parameter space of relevance to contrails is very large (ambient conditions, aircraft-dependent conditions, aerosol content, etc.), only a tiny select fraction of which is sampled in the simulations here. Accordingly, some of the statements of results in the paper are stated too strongly or too broadly and need added qualifications so as not to mislead some readers: they may be true for the specific simulations conducted, but that doesn't mean they hold across the full parameter space (and in some cases definitely do not). Examples include the statements in lines 16-17, 456-457, 467-468, 476-477, and 600-602.*

Of course, we do not cover the complete parameter space within this manuscript and it is clear that one has to take care of the restricted parameter variation when drawing conclusions. Nevertheless, we tried to advance and to extend the variation of important sensitivity parameters opposite to recent studies. While most recent studies focused on soot number emission indices, we included further specific properties as the hygroscopicity parameter as well as mean dry core size and width of the size distribution.

Furthermore, we tried to generalise the impact on ambient conditions by introducing Delta_T (difference between ambient and SA-threshold temperature) that can vary with ambient temperature, pressure and relative humidity over ice. We predominantly find high sensitivity of contrail ice crystal numbers to atmospheric parameters and certain soot properties near the contrail formation threshold consistent with previous box model studies or analytical approaches (e.g., Kärcher & Yu, 2009; Kärcher et al., 2015; Bier & Burkhardt, 2019).

We weakened the statements in the examples you listed. Either by removing the word "only" or by adding "in our model set-up" or rather "for our parameter settings". Moreover, we replaced the original statement in the abstract/conclusions "Absolute ice crystal numbers are, on the other hand, controlled by the soot number emission index for all ambient conditions" by

„[...]The freezing fraction displays a slight decrease with increasing soot number emission index, particularly for higher soot emission levels. This weakens the increase of absolute ice crystal numbers with increasing soot number emission index."

*(8) The authors are likely overstating the importance of the ``deactivation phenomenon'' they identify in lines 393-401 in explaining reductions in freezing fraction. While such reductions are almost certainly due to competition for available moisture between different aerosol populations (as discussed in Lew20), ``deactivation'' is only one such mechanism involved. The alternate mechanism of competition between aerosol for moisture preventing some aerosol from ever activating (rather than activating and then deactivating) is generally more responsible for reductions in ice number (judging from the larger parametric study of Lew20).*

You are right that the subsequent activation of exhaust particles, where larger/more hygroscopic particles form droplets first and may prevent the droplet formation on smaller/less hygroscopic particles, is of higher importance than the deactivation of already activated droplets. However, this deactivation phenomenon makes up, particularly for our high soot cases, significant differences in final ice crystal numbers between both averaging modes and deserves at least a reference in the discussion section. Thereby, we argue that the weak dilution of the original FLUDILES data is conducive to excessive deactivation in the average traj set-up and would be of lower relevance for the accelerated trajectories. Moreover, we do not mention this deactivation phenomenon in the conclusions so that its importance is in our opinion not overstated in the manuscript.

Minor points:

*(9) line 40: Contrary to what is implied in the statement, large enough supersaturation to form ice crystals starting from ultrafine aerosol can be produced in jet exhausts fairly easily (as is shown in Lew20).*

Brock et al. (2000) show that the measured number distributions of volatile particles are dominated by radii lower than 2.5 nm displaying peaks of around 1.5 nm. Due to the Kelvin effect, these small (so called ultrafine) particles typically require water supersaturation of at least more than 10%. For instance, 1.5 and 2.5 nm sulfuric acid particles (assuming hygroscopicity parameter of 0.9) need water supersaturations of 12% and 27%, respectively, to activate into water droplets.

We have replaced "huge" by "high (> 10%)" in the introduction to relativize our original statement.

*(10) lines 132-134: The statement is misleading. Just as one can include a soot spectrum without necessarily modeling its formation, one can usefully include ultrafine aerosol without modeling its formation from ion clusters (as done, e.g., in Lew20).*

You are right. It would have been possible to include a size spectrum of ultrafine volatile particles without modelling its formation. We have deleted that statement from section 2.2.1 and added following lines instead:

"Even though ultrafine volatile particles become more substantial for contrail ice crystal formation in a 3D LES set-up (Lewellen, 2020), they are supposed to have a tiny impact on contrail formation for current soot-rich emissions in a box model framework (Kärcher & Yu, 2009). Hence, we neglect volatile particles in the present study, but we will incorporate this particle type in the next model version."

As stated in the introduction, our focus was to investigate the sensitivity of contrail ice nucleation to a broad range of different soot properties. The other particle types like ambient and ultrafine volatile particles will be considered in future studies.

*(11) Some of the figure captions need additions to make the figures more understandable on their own. For example some of the description following line 340 should be moved from the text to the figure caption. And the red and blue lines should be identified in the fig.4 caption.*

Thank you for the suggestions. We have moved some technical description at the beginning of section 3.1 to the Fig. 1 caption. The red and blue lines are now described in Fig. 4 caption. Moreover, we clarified the adaption of *EIs* for the last row of Fig. 5 by adding that *EIs* is changed proportional to r_d^-3 to keep the total soot mass constant.

*(12) The wording in lines 351 and 354 is somewhat misleading. While RH_liq increases with time and radius within the bounds of fig 1 it does not do so indefinitely: for large enough radius and time it decreases.*

Indeed, this was not well-formulated. From the Schmidt-Appleman-Plots (Schumann, 1996) we know that plume supersaturation is a transient phenomenon. We have clarified that the increase of hypothetical RH_liq with time only occurs at smaller plume age and radial distance and that this dependency of RH_liq reverses for higher t and r.

*(13) line 446-447 is misleading. Since the ambient is supersaturated in the case illustrated in fig.4, conditions don't drop back to ice saturation and the crystal size continues to grow (albeit more slowly).*

Of course, ambient conditions do not drop back to ice-saturation but the plume becomes ice-saturated after a certain time. Therefore, we added

"The curves level off as soon as ice-saturation is reached **in the plume**".

*(14) In line 530, differences due to different turbulence realizations in the Lew20 LES results are described as an ``irreducible'' uncertainty. This is not strictly true: one could average over an ensemble of such LES results to reduce this uncertainty (at additional numerical cost).*

Clearly, one can average over an ensemble of LES to obtain a mean evolution. But this is not the point we wanted to make. What we wanted to express with "irreducible" that even for identical parameters and same statistics of the turbulent fluctuations the contrail properties show some spread due to turbulence. The original statement stresses this probabilistic aspect and the non-deterministic behaviour.

**References that do not appear in the manuscript**

Brock, C. A., Schröder, F. P., Kärcher, B., Petzold, A., Busen, R., Fiebig, M., and Wilson; J. C. (2000). Ultrafine particle size distributions measured in aircraft exhaust plumes, J. Geophys. Res., 105, 26, 555–26, 568.

Grabowski, W., W., H. Morrison, S.-I. Shima, G. C. Abade, P. Dziekan, und H. Pawlowska. (2019). Modeling of Cloud Microphysics: Can We Do Better? Bull. Am. Meteorol. Soc., 100(4): 655-672, 20. doi:10.1175/BAMS-D-18-0005.1

Pope, S. B. (2000). Turbulent flows, Cambridge University Press.

Unterstrasser, S., F. Hoffmann, und M. Lerch. (2017c). Collection/aggregation algorithms in Lagrangian cloud microphysical models: rigorous evaluation in box model simulations. Geosci. Model Dev., 10(4):1521-1548. doi: 10.5194/gmd-10-1521-2017

---

## Author Comment (AC2)

Dear Bernd,

please find our answers to your comments below (your comments are repeated in italics).

*Aircraft exhaust plume turbulence affects the formation and properties of contrail ice crystals and how predictions of nucleated ice numbers and sublimation losses relate to aircraft measurements [Kärcher, 2018]. This manuscript draft describes a project activity relating to the question how contrail ice formation might be affected by coupling plume turbulence and ice microphysics.*

*Decades of research established the basic contrail ice formation pathway (activation of size-dispersed plume and ambient aerosols present in decaying jet aircraft exhaust plumes). The most important findings, used as the basis of an intentionally simplified parameterization scheme [Kärcher et al., 2015], have been confirmed by field measurements. This is particularly true, on a quantitative basis, for the number of contrail ice crystals as a function of aircraft-related parameters and ambient conditions.*

*While I understand the desire to ultimately include more microphysical complexity into LES in the author's quest for gaining new insights, open research issues that potentially challenge established findings should be clearly identified and formulated. Here I mean those issues that we have incomplete knowledge of or that contradict observations. In my view, the authors could improve on this, especially in the light of their concluding statement (line 620): "Hence, using a large spatially resolving trajectory ensemble does not necessarily lead to improved scientific results contrary to what we expected in the beginning." and the significant progress in coupling turbulence and microphysics in 3D-LES reported by Lewellen [2020].*

Most results and findings you refer to were obtained with simple box models. With regard to model development, the paper by Lewellen (2020) is a significant progress. For instance, Lewellen (2020) found that ice crystal formation on ultrafine volatile particles is more substantial in 3D LES than in a box model updating earlier findings in Kärcher & Yu (2009). In our opinion, it is not necessary to hypothesize in advance on established findings that are potentially challenged by a new model.
Our goal is to setup a modelling system that is similarly-tailored as the model in Lewellen (2020) and which allows similar, yet independent simulation studies.
As 3D simulations of contrail simulations are costly, an alternative less demanding approach would be also desirable. In this regard, the present manuscript investigates the approach of applying a box model for a large trajectory ensemble. We conclude that independent computations without any information exchange among the various trajectories give unsatisfactory results.
This issue is discussed in the evaluation section and corresponding remarks are also given in the next-to-last paragraph of the conclusions, but statements in the abstract were still missing. Therefore, we have added
"Using an ensemble mean framework instead of a single trajectory does not necessarily lead to an improved scientific outcome. Contrail ice crystal numbers tend to be overestimated since the interaction between the different trajectories is not considered."

at the end of the abstract.

*In their project, the authors opted to include an intermediate-complexity microphysical approach into their framework, which basically replicates the original parameterization approach [Kärcher et al., 2015]. For example, neglecting the liquid phase denies the opportunity for further in-depth study or sanity checks on older results. This seems particularly relevant, as the author's ultimate goal is to include the described methodology in 3D-LES (line 23f). In a more realistic setting, consideration of a kinetic description of droplet activation and ice nucleation is arguably required for proper simulations of the interaction between droplet and ice microphysics and turbulent entrainment-mixing in plume regions, where all these processes develop on similar time scales.*

It is true that our microphysical pathway for contrail formation is very similar to Kärcher et al. (2015), but there are strong conceptual differences of how the microphysical processes are modelled: Kärcher et al. (2015) assume within their analytical approach that all droplets form on soot or entrained ambient particles at one particular time in the fast cooling plume. Thereby, that droplet number concentration that is needed to balance a further increase of plume supersaturation due to condensation loss is determined. This is of course a simplification since important competition effects like subsequent activation of different particle type/sizes and the interaction of exhaust particles with plume relative humidity are not considered.

In our study, the box model explicitly simulates the contrail formation process in a simple dynamical framework. In our particle-based microphysical approach, soot particles and hydrometeors (i.e., ice crystals or water droplets) are described by simulation particles (SIPs). Each SIP stores information about the phase, dry and wet radius and weighting factor.

In the 2$^{nd}$ sentence, you imply that we would neglect the liquid transition phase which is not true. Soot particles first activate into water droplets (if water supersaturation is sufficiently large) and subsequently freeze to ice crystals. Opposite to Kärcher et al. (2015) droplet and ice crystal formation occurs at different plume ages (depending on the particle dry core size) and the hydrometeors directly interact with plume water vapour and temperature (latent heating) at least within one trajectory. Using the diffusional growth equations, our droplet activation automatically exhibits a kinetic character. The homogeneous ice nucleation indeed occurs instantaneously assuming that the entire droplet freezes immediately once ice nucleates within its volume. But this is a reasonable approximation since ice crystal growth in contrails is slow relative to droplet freezing (Ford, 1998).

*Explicitly simulating water activation of liquid or mixed-phase aerosols (here: exhaust soot coagulating with the evolving ultrafine aqueous aerosols) alongside homogeneous freezing is pretty standard in cloud physics. Numerical representations are available that are on the one hand consistent with the original LCM treatment of aerosol and ice growth [Sölch & Kärcher, 2010] and on the other hand employed, with an even greater level of complexity than needed for contrail studies, even in global climate models [Jacobson, 2002].*

*The intercomparison of results with other models, be it an LES or a high-complexity microphysical model, is clearly meaningful, especially when measurements are difficult to interpret. However, in this case, airborne measurements of the temperature and humidity dependence of contrail formation [Bräuer et al., 2021] are available to put the model predictions to the test. The comparison of the author's extended approach with the original parameterization [Kärcher et al., 2015] is less valuable for validation, since they base most of their methodology (Section 2.2) on this parameterization.*

The study of Bräuer et al. (2021) appeared on 28$^{th}$ April and we submitted our manuscript on 29$^{th}$ April. Hence, we were not able to include this study in our work.

Clearly, a validation effort must include a comparison with observational data in order to check if the assumption on the underlying model physics are well-chosen. But this is only one aspect. The other point is that the numerical implementation of solving the underlying model physics should do what it is supposed to do (also called "verification" in software design).

As stated above, the underlying model physics in the present study and in Kärcher et al. (2015) are similar, yet the approaches to solve the model physics are quite different. Hence, we do not agree that a comparison with Kärcher et al. (2015) would be less valuable. It is interesting to see results of different methodologies that are based on similar model physics.

*I have a number of further points the authors may wish to clarify/reassess/explain/check/update/expand upon.*

*38-41: Ultrafine aqueous plume particles have been shown to form a second contrail ice mode from uptake of nitric acid to form ternary H2O/H2SO4/HNO3 solutions, partial activation and homogeneous freezing alongside water activation [Kärcher, 1996].*

This is an interesting aspect, but Fig. 1 of Kärcher (1996) clearly shows that the number concentrations of the second volatile particle mode (H2SO4/H2O) and the HNO3/H20 solution droplets mode are several orders of magnitude lower than the number concentrations of the first volatile particle and the soot particles mode. Although the larger volatile particles in the second mode are easier to activate than the smaller ones in the first mode, their contribution to contrail ice crystal formation is very low (see also argumentation in the next answer.)

*40: The large size tail of number-size distributions of ultrafine aqueous plume particles is extremely steep [Brock et al., 2000], so the supersaturation needed to water-activate these particles is highly variable and includes values that barely exceed liquid water saturation.*

Even though measured volume and surface area display a second mode for particles with diameter (D) larger than 10 nm, the number distribution of volatile particles is clearly dominated for D < 5 nm with maxima at around 3 nm (Brock et al., 2000). These small particles require, due to the Kelvin effect, high water supersaturations (e.g., 25% for 3 nm H2SO4 particles using Kappa-Köhler-Eq.) to form water droplets. Of course, the larger particles (D > 10 nm) at the distribution tail are much easier to activate (critical supersaturation of a few percent), but their contribution to the overall volatile particle spectrum is very low.

The box model study of Kärcher & Yu (2009) shows that ultrafine volatile particles have only a significant impact on contrail ice nucleation for soot-poor emissions and ambient temperatures of at least more than 5 K below the formation threshold while their impact is very low for soot-rich emissions (as they typically occur behind commercial aircraft). The latter is because the droplet formation on the particularly larger soot particles and the corresponding depletion of plume humidity typically suppresses the activation of ultrafine volatile particles even at lower ambient temperatures.

In contrast, the 3D LES study of Lewellen (2020) indicates that a significant fraction of contrail ice crystals originates on ultrafine particles even in combination with high soot emissions (see his Fig. 3). This means that the relevance of exhaust or entrained ambient particles on contrail formation might change significantly when switching from a 0D box model set-up to 3D LES.

*41: Can you be more specific what you mean by seconds? This assertion requires evidence. In which conditions away from the formation threshold would it take longer than 0.5–1 s to form contrail ice?*

We have replaced "in the first second(s)" by **"within the first second"**.

This is valid at ambient temperatures (Ta) far away from the formation threshold. Only close to the formation threshold (Delta_T <~ 1.5 K), ice crystal formation might take longer than 1 s as displayed in our Fig. 2d. Note that the latter results holds for the original FLUDILES trajectories that suffer from the time lag due to the neglection of the conversion of jet kinetic in thermal energy. If these trajectories are accelerated (explained in section 5 and shown in Fig. 7), contrail ice crystals indeed form within the first second even at conditions very close (Delta_T < 0.3 K) to the formation threshold.

*52-54: Small soot particles will not water-activate in threshold conditions, so this argument seems to be moot. What is the sensitivity of threshold ice numbers (then originating from the largest soot particles) on the Kelvin effect and what is the uncertainty in determining the underlying surface tension? Assuming water saturation to be sufficient for soot activation has only been used as a reasonable approximation in the contrail parameterization [Kärcher et al., 2015]; the underlying numerical process model does not make this assumption.*

First, it is not true that Kärcher et al. (2015) assume water saturation to be sufficient for the activation of soot particles. (This aspect is explained in more detail in the answer to the question regarding the "adapted version of the parameterization".)

Fig. 1 B (Appendix of the current manuscript) displays that the critical saturation ratio for activation of soot particles ($Sc$) increases with decreasing dry core radius r$dry$. This increase becomes significantly stronger for *rdry* below 15 nm due to the non-linear Kelvin term. The smallest soot particles (*rdry* < 10 nm) require high plume water supersaturation (> 20%) which is not achieved at near-threshold conditions. Therefore, those soot particles cannot activate into water droplets and subsequently freeze to ice crystals (excluding heterogeneous ice nucleation). We have replaced "small" by "only several nanometer sized" for clarification. In this study, we assume the surface tension of pure water droplets that includes a temperature dependency based on experimental findings (see also answer to your last point).

*56-60: Why "However"? This is not (necessarily) a contradiction.*

This is true. We replaced "in general leading to a hydrophobic character" by "causing a weak hygroscopicity" and removed "However" in the subsequent sentence.

*100: In my opinion, such exhaust soot particle properties are less uncertain than claimed in line 133ff [Moore et al., 2017].*

Experimental data of measured soot properties are quite limited both at ground level and even more at cruise altitude conditions. Hence, there is of course a significant uncertainty. This point is also discussed in section 3.1. of Kärcher et al. (2015):

"Number EIs for aircraft soot particles can be inferred from mass EIs using fixed mass-size relationships [...] Predicting EI$_s$ from equation (7) requires accurate *rs* [median dry core radius] values. Variations in *rs* and M$s$ [soot mass emission index] alone cause in-flight soot number emission indices to lie between $10^{14}$ and $10^{15}$ (kg-fuel)$^{-1}$. **However, experimental data sets suitable for constraining mass-size relationships for cruising conditions are very limited.**"

*144: While numerical results may converge, I wonder about the spatial resolution of the ice crystal mode. I understand that contrail ice is resolved by 50-200 SIPs, yet typically tens of thousands of ice*

*crystals form in contrails per cubic centimeter of air. How many real ice crystals are represented by one SIP on average? Can you estimate how many SIPs will be needed in the full 3D set-up in order to obtain a reasonable spatial coverage across the entire plume cross-section?*

Let us define $n_{weight}$ as the number of real ice crystals that are represented in one SIP. The average ice crystal number $n_{weight,m}$ can be derived from the trivial relationship $n_{weight,m} = N_{IC}/N_{SIP}$, where $N_{IC}$ and $N_{SIP}$ are the total numbers of ice crystals and SIPs, respectively. The total ice crystal number depends on the soot number emission index and the fraction of soot particles freezing to ice crystals. In our baseline case, $n_{weight,m}$ is around $10^{10}$ for a single average trajectory. In the ensemble runs with 25000 trajectories sampling the expanding plume, $n_{weight,m}$ is accordingly $N_{IC}/(25000*N_{SIP})$.

Note that the quality of a particle-based microphysics simulation cannot be judged by means of the $n_{weight}$ value. A large absolute value of $n_{weight}$ does not imply that a specific simulation is coarse. Analogously, one would not change the number of bins in a bin model depending on the ice crystal number in a cloud. An adequate number of SIP or bins depends much more on the dispersion of the ice crystal size distribution and other factors, but clearly not on $n_{weight}$.

At the current stage, it makes no sense to estimate the number of SIPs required in 3D simulations of contrail formation. We will make corresponding convergence tests once the 3D setup is established. By the way, Unterstrasser & Sölch (2014) found that the number of required SIPs per grid box is smaller in higher-dimensional domains due to averaging effects. We expect the same benefits from averaging in the upcoming simulations. Moreover, the LCM code is equipped with SIP merging and splitting operation to interactively adapt the number of SIPs. The splitting operation will be useful in strongly expanding exhaust jets/plumes which enables us to increase the number of SIPs over time.

*165: How sensitive are the critical supersaturations (and derived variables, ultimately, the contrail ice numbers) calculated based on eq 1 to uncertainties in surface tension? In understand that the method does not track the acid or water mass fractions in the soot particle coatings deviating from the high-complexity models (and therefore also keeps the parameter kappa in eq.1 constant). How then is the surface tension of the acidic solutions estimated?*

In this study, we assume the surface tension of **pure water** droplets. Acids in the soot particle coating can be positively absorbed by the water droplet, where "positive absorption" means that acid molecules concentrate near the droplet surface. This causes a decrease in surface tension relative to that for pure water droplets. However, the volume fraction of soot coating substances (like sulfuric acid) makes up only a few percent of the overall soot particle/droplet volume (e.g., Petzold et al., 2005). Therefore, the impact of soot particles coating on surface tension and, accordingly, on critical saturation ratios are supposed to be negligible and we do not consider the variation of droplet surface tension with acidic solution concentrations.

This is now clarified in the newly created Appendix B1) (see also the answer to your last point).

*186: Why would the method to estimate the freezing threshold temperature be suitable only in "strong cooling situations" and why don't the author's refrain from basing their estimates of freezing fractions on actual freezing rates? The latter contain valuable kinetic information. In doing so identical to the original parameterization [Kärcher et al., 2015], this approach tends to maximize freezing fractions.*

Concerning your first question our wording was misleading. The method is not "only" but "**also"** suitable in strong cooling situations. We have clarified the sentence by writing "This method **can handle a** strong cooling situation …".

We do not agree with the last point since instantaneous freezing of water droplets in exhaust plumes is a reasonable assumption as explained above.

*584: What is an "adapted version"? if changes have been made to the original parameterization [Kärcher et al., 2015], the impact of these changes on ice crystal numbers should be documented, as the performance of the original parameterization was tested against observations.*

Kärcher et al. (2015) use the Kappa-Köhler theory (Petters and Kreidenweis, 2007, abbreviated hereafter by PK2007) to calculate the so called activation dry core radius for soot ($r_{d,act,s}$) and entrained ambient particles ($r_{d,act,a}$) for given plume water saturation ratio $S_p$. This means that all soot (ambient) particles with dry core radii $r_{d,s} >= r_{d,act,s}$ ($r_{d,a} >= r_{d,act,a}$) will activate into water droplets if a certain water supersaturation is reached. Note that in this method, $S_p$ is effectively considered as the critical saturation ratio $S_c$ (maximum of the Köhler curve) and $r_{d,act}$ is calculated as the corresponding dry core radius to $S_c$. Kärcher et al. (2015) **prescribe the hygroscopicity (kappa) of soot with 0.005** and set the solubility of ambient particles to 0.5. For the calculation of $r_{d,act}$ Kärcher et al. (2015) use the **simplified fit Eq. (10) of PK2007** (even including a transcription error which has been corrected in Bier & Burkhardt, 2019), which is only valid for **kappa >~ 0.1**. While this equation is appropriate to treat ambient particles, it is not valid for soot particles due to their weak hygroscopicity. The Figure below displays our calculated $S_c$ for the activation of soot particles (applying the method described in App. B2) at different plume temperatures, where typically droplet formation occurs, and that obtained from Eq. (10) of PK2007. We see that the latter significantly overestimates $S_c$ for dry core radii < 15 nm.

[Figure]

Critical saturation ratio ($S_c$) for activation of soot particles (setting the hygroscopicity parameter to 0.005) depending on their dry core radii for different plume temperatures (T) calculated as the maximum of the Kappa-Köhler Equation (6) from Petters & Kreidenweis (2007) by applying Newtonian Iteration. The green dashed line shows the result for $S_c$ when using the simplified Equation (10) from Petters & Kreidenweis (2007).

In the adapted version of Kärcher et al. (2015), we replace the simplified fit Eq. (10) of PK2007 for soot particles by a polynomial fit of the black line (in the Figure) for a plume temperature of 236 K. For the reasons described above, we favour to use that adapted version in our study. Testing the **original** parameterisation against observations is no reason to suppress improvements or corrections in microphysical implementations of that parameterisation.

*602f: The maximum supersaturation is controlled by the contrail ice crystal number concentration.*

Of course, the (maximum) plume water supersaturation is also controlled by the droplets/ice crystals that form on soot particles. In the sentence, the hypothetical supersaturation that purely results from the thermodynamic formation criterion (in absence of microphysics) was meant.

Therefore, we have added "(**in terms of pure thermodynamics**)" behind "maximum supersaturation" for clarification.

*604ff: Please explain why absolute ice numbers are only sensitive to the total soot number while freezing particle fractions are only sensitive to soot particle size and solubility. At current soot emission levels, contrail ice formation is limited by the plume cooling rate. It would be interesting to know at which soot emission levels contrail ice formation will be limited by the availability of soot particles at emission for given ambient temperature.*

First, there seems to be a misunderstanding. Since the apparent ice number emission index ($AEI_{ice}$) is the product of the freezing fraction and soot number emission index ($EIs$), absolute ice numbers are just as sensitive to average soot particle size and solubility like the freezing fraction (for fixed $EIs$). Our original concluding sentence "Absolute ice crystal numbers are, on the other hand, controlled by the soot number emission index for all ambient conditions" is simply based on the linear relationship between $AEI_{ice}$ and $EIs$, which is actually trivial. Therefore, we have replaced our statement in Sect. 6 by

„[...]The freezing fraction displays a slight decrease with increasing soot number emission index, particularly for higher soot emission levels. This weakens the increase of absolute ice crystal numbers with increasing soot number emission index."

and accordingly, we have modified the wording in the abstract (lines 19-20).

*Appendix A: How accurate is eq A2 at plume temperatures well in excess of ambient air temperature? How important are latent heat effects?*

Thermal conductivity is only used in the diffusional growth equations and, therefore, is relevant at temperatures below 250 K when the first droplets form. Hence this physical quantity is not at all a crucial parameter in the temperature range you mention. Plume temperature changes due to latent heat have a magnitude of around $10^{-4}$ to $10^{-3}$ K per time step and, therefore, they are of low importance.

*Appendix B: How is sigma in eq B1 calculated and how well is it known? See also comments above (l52ff and l165).*

For the liquid phase, we assume the surface tension of pure water droplets. It is calculated according to Pruppacher & Klett (1997) using a temperature dependent fit formula based on experimental data from Hacker (1951). For the ice phase, we prescribe a fixed value (Pruppacher & Klett, 1997).

We have now documented the calculation of the surface tension in Appendix B.

**References that did not appear in the manuscripts**

Bräuer, T., Voigt, C., Sauer, D., Kaufmann, S., Hahn, V., Scheibe, M., et al. (2021). Airborne measurements of contrail ice properties—Dependence on temperature and humidity. *Geophysical Research Letters*, *48*, e2020GL092166.

Brock, C. A., Schröder, F. P., Kärcher, B., Petzold, A., Busen, R., Fiebig, M., and Wilson; J. C. (2000). Ultrafine particle size distributions measured in aircraft exhaust plumes, *J. Geophys. Res.*, *105*, 26, 555–26, 568.

Ford, I. J. (1998). Ice nucleation in jet aircraft exhaust plumes, in *Air Pollution Research Report 68: Pollution From Aircraft Emissions in the North Atlantic Flight Corridor (POLINAT 2), Report EUR 18877*, edited by U. Schumann, pp. 269–287, European Commission DG, Brussels, Belgium.

Hacker, P. T. (1951). Experimental Values of the Surface Tension of Supercooled Water, Technical Note 2510, National Advisory.

Kärcher, B. (1996). Aircraft-generated aerosols and visible contrails, *Geophys. Res. Lett*., *23*(15), 1933-1936.

Petzold, A., M. Gysel, X. Vancassel, R. Hitzenberger, H. Puxbaum, S. Vrochticky, E. Weingartner, U. Baltensperger, and P. Mirabel (2005), On the effects of organic matter and sulphur-containing compounds on the CCN activation of combustion particles, *Atmos. Chem. Phys.*, *5*, 3187–3203.

---

## Author Response (AR2)

**Reply to Editor Decision and the two included Reviewer Comments**

The Editors Decisions and the reviewers' comments are repeated in italics. Our response is added in bold font and green colour.

*Dear Andreas Bier and co-authors,*

*I am pleased to let you know that. based on the two reviews of the revised manuscript, your manuscript has now been accepted for publication in ACP after minor revsions (review by editor).*

*Both referees have final remarks to the paper (listed below) which I would ask you to consider and answer before uploading the final manuscript.*

*Kind regards, Martina Krämer (as CP Senior Editor)*

**Dear Martina Krämer,**

**many thanks for your positive evaluation of our manuscript revision. In the following we reply to the few reviewer comments. In addition to the modifications based on our replies, we have corrected the units in Eq. (4) of the manuscript.**
* * *
Referee #1:
*With their addition of some clarifications, caveats, and promises for improvements in future work, the authors have in this revised version adequately addressed the concerns listed in my review of the original manuscript, at least sufficiently for its role as a ``model development'' paper. With regards to the author's replies to my prior comments I have three further suggestions, the first for the present manuscript, the others directed more toward their future work:*

*(R1) In their response to my prior comments the authors provided a definition of the plume radius $r\_p$ appearing in the paper as ``the distance of the most remote trajectory from the plume centre at a given plume age'', but (unless I missed it somewhere) a definition has not been included in the revised manuscript itself. For completeness one really should be.*

You are right. We have included that definition of $r\_p$ at the beginning of Section 3.1

*(R2) The authors state that they have diffused the expected sharp gradient at the plume edge in the initial conditions to the smoothed profile shown in fig.1 in order to avoid numerical instabilities in their model. The unphysical effects of this are of lesser concern for box model applications as in the present manuscript, but will be more important in the full LES mode that the authors are aiming toward. Accordingly I strongly encourage the authors to put in some work on their numerics to avoid this in their future efforts. My answer, by the way, to the query in their comments: ``We wonder if you do not have to include such precautionary measures in your model...'', is a firm ``no''.*

The trajectory data stem from the FLUDILES code. The ice microphysics code LCM, which we extended to cover contrail formation in the present study, is however coupled to the LES code EULAG. The numerical solution technique for advective transport differs between those two codes We will pay attention to this aspect within the EULAG-LCM set-up.

*(R3) With regards to the question of binned versus particle-based microphysics and the resolution requirements of each: I did not state a belief in my original review that binned methods were in general superior to particle-based ones, as the authors seem to have inferred; each have their own*

*strengths and weaknesses. Rather I was suggesting that with the bin choices made in Lew20 and the choice of SIPs made in the present work that the former likely resolves ice and droplet spectra more faithfully. One useful set of tests I have performed in trying to determine resolution requirements for the microphysics in my own model (in addition to standard sensitivity tests) is to compare the results of different choices with exact analytic solutions where possible. In Lewellen 2012 (``Analytic solutions for evolving size distributions of spherical crystals or droplets undergoing diffusional growth in different regimes'', DOI: 10.1175/JAS-D-11-029.1), exact analytic solutions are derived that are of direct relevance to contrails. Appendix B of that paper illustrates the numerical requirements for binned microphysics schemes to match various aspects of the exact solutions at different levels of accuracy. The same exercise could be performed for particle based methods. I strongly encourage the authors to explore such tests for their planned LES work.*

Even though we believe that our convergence tests (varying the number of simulation particles) are sufficient, comparison to the analytical solution of your specific test case is indeed a nice benchmark case and can be implemented to verify several components of the LCM box model.

However, we want to mention that particle-based methods often contain a probabilistic component in multi-dimensional set-ups. This implies that relatively few simulation particles in every grid box can suffice for a reliable prediction of total quantities. Only if one is interested in local quantities, (e.g., deriving a PDF of number concentration) the requirements on SIP numbers are higher. For a deeper understanding of those aspects, we refer to Unterstrasser & Sölch (2014), Unterstrasser et al (2017) and Unterstrasser et al (2020).
* * *
Referee #2:
*The authors have responded to the reviewer comments in great detail and the paper is improved as a result.*
*I have only one further suggestion, which is to include a discussion (in Conclusions) on experimental data/evidence that would improve the accuracy of model outputs and/or allow the model to be applied more widely. For example, are there chemical and physical properties of soot particles that should be better characterised, experimental characterisation of plume dilution, meteorological observations, properties of aircraft and their engines that are not publicly disclosed. This could certainly be the topic of another paper but here I suggest that a short discussion is included to help motivate/inform/influence experimental research. After all, the authors have an insight into these important properties/parameters that would be extremely valuable to share.*

Thank you for your positive feedback. Regarding your suggestion we have added following lines at the end of the Conclusions:
"**To improve the accuracy of model outputs and to enable a better evaluation with in-situ campaign data, contrail formation experiments should include in addition to total number concentrations measurements of exhaust and ice particle size distributions. Due to the large spatial heterogeneity and fast change of thermodynamic and microphysical quantities, a precise determination of contrail age and sampling position in the plume is crucial. High resolution measurements of water vapour can help to characterise the growth and sublimation of ice crystals. A thorough discussion of benefits and limitations of different measurement strategies are valuable to understand how a quantitative comparison of modelled and measured data should be performed.**"

**References**

Unterstrasser, S. and Sölch, I.: Optimisation of simulation particle number in a Lagrangian ice microphysical model, Geosci. Model Dev., 7, 695-709, doi:10.5194/gmd-7-695-2014, 2014.

Unterstrasser, S., F. Hoffmann, und M. Lerch. (2017). Collection/aggregation algorithms in Lagrangian cloud microphysical models: rigorous evaluation in box model simulations. Geosci. Model Dev., 10(4):1521-1548. doi: 10.5194/gmd-10-1521-2017

Unterstrasser, S., Hoffmann, F., and Lerch, M., 2020: Collisional growth in a particle-based cloud microphysical model: insights from column model simulations using LCM1D (v1.0), Geosci. Model Dev., 13, 5119-5145, doi:10.5194/gmd-13-5119-2020